# EDT: An Efficient Diffusion Transformer Framework Inspired by Human-like Sketching

**Xinwang Chen**[*1], **Ning Liu**[*1], **Yichen Zhu**[1], **Feifei Feng**[1], **Jian Tang**[†2]
[1] Midea Group, [2] Beijing Innovation Center of Humanoid Robotics
chen_xinwang@xs.ustb.edu.cn, ningliu1220@gmail.com
{zhuyc25, feifei.feng}@midea.com, jian.tang@x-humanoid.com

## Abstract

Transformer-based Diffusion Probabilistic Models (DPMs) have shown more potential than CNN-based DPMs, yet their extensive computational requirements hinder widespread practical applications. To reduce the computation budget of transformer-based DPMs, this work proposes the **E**fficient **D**iffusion **T**ransformer (EDT) framework. The framework includes a lightweight-design diffusion model architecture, and a training-free Attention Modulation Matrix and its alternation arrangement in EDT inspired by human-like sketching. Additionally, we propose a token relation-enhanced masking training strategy tailored explicitly for EDT to augment its token relation learning capability. Our extensive experiments demonstrate the efficacy of EDT. The EDT framework reduces training and inference costs and surpasses existing transformer-based diffusion models in image synthesis performance, thereby achieving a significant overall enhancement. With lower FID, EDT-S, EDT-B, and EDT-XL attained speed-ups of 3.93x, 2.84x, and 1.92x respectively in the training phase, and 2.29x, 2.29x, and 2.22x respectively in inference, compared to the corresponding sizes of MDTv2. Our code is available at here.

## 1 Introduction

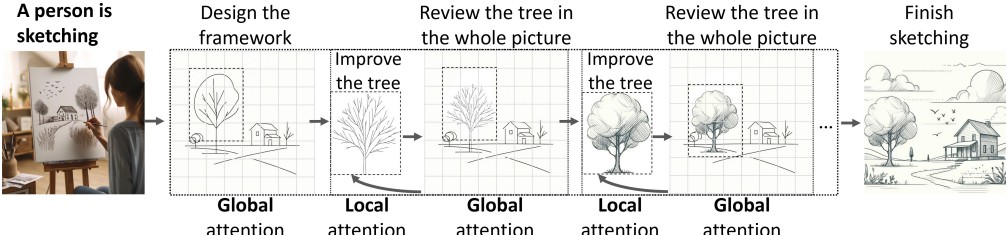

Figure 1: Illustration of the alternation process of local and global attention during sketching.

Numerous studies [1, 2, 3, 4, 5] and practical applications [6, 7, 8] have validated the effectiveness of Diffusion Probabilistic Models (DPMs), establishing them as a mainstream method in image generation. In past years, predominant works [1, 2, 3, 4, 9, 10, 11] have advanced diffusion models by incorporating a convolutional UNet-like [12] architecture as their backbone. On the other hand, transformers [13] have achieved significant milestones in both natural language processing [14,

---

[*]Joint first authorship. Either author can be cited first.

[†]Corresponding author.

38th Conference on Neural Information Processing Systems (NeurIPS 2024).

15] and computer vision [16, 17, 18, 19], prompting recent attempts to integrate these powerful transformer-based architectures into diffusion models with considerable success. For instance, U-ViT [20], an early work in diffusion leveraging ViT-based transformers, surpassed the contemporary CNN-based U-Net DPMs in class-conditional image generation on ImageNet, demonstrating their potential. Similarly, diffusion transformer (DiT) [21], which employs transformers as its backbone instead of the traditional U-Net backbone in latent diffusion models (LDM) [11], has shown excellent scalability. Further, masked diffusion transformer (MDT) [22] observes that DPMs often struggle to learn the relations among object parts in an image. To solve this, MDT introduces a masking training scheme to enhance the DPMs' ability to relation learning among object semantic parts in an image. MDT established a SOTA of class-condition image synthesis on the ImageNet.

While transformer-based DPMs offer scalability and a higher performance ceiling than their CNN counterparts, they also require more computational resources. For instance, in each inference step, DiT-XL-2 consumes 118 GFLOPs and U-ViT-H requires 133 GFLOPs. This computational demand escalates with increasing time steps or token length, limiting their practical application. Despite their computational inefficiency, few studies have explored enhancing the efficiency of transformer-based DPMs. Therefore, the trade-off between computation and performance underscores the importance of designing a lightweight model architecture that maintains excellent performance.

To improve the computational efficiency of transformers in DPMs, we introduced a comprehensive optimization framework named **E**fficient **D**iffusion **T**ransformer (EDT). Specifically, we developed a lightweight diffusion transformer architecture based on a comprehensive computation analysis. Moreover, we devised the Attention Modulation Matrix (AMM) and its alternation arrangement in EDT inspired by human-like sketching. AMM, functioning as a plug-in, can be seamlessly integrated into diffusion transformers to enhance image synthesis performance significantly without requiring additional training. Additionally, we introduced a novel token relation-enhanced masking training strategy tailored for EDT to enhance its relation learning capability.

**Lightweight-design diffusion transformer** Based on the empirical analysis of the number of tokens, token dimensions, and the FLOPs, we propose two principles to design the lightweight diffusion transformer, and redesign and incorporate the down-sampling, up-sampling, and long skip connection modules into diffusion transformers. The utilization of down-sampling module can reduce FLOPs, but harms performance, since the token merging operation in down-sampling and long skip connections modules leads to the loss of token information. To mitigate this loss, we enhance the key features by introducing token information enhancement and positional encoding supplement.

**Attention Modulation Matrix** The mind stores visual structures as a top-down hierarchy passing from general shape to the relationships between parts down to the detailed features of individual parts [23, 24]. Based on this storage structure in the mind, humans tend to follow a coarse-to-fine drawing strategy [25]. The logical structure of sketching of humans tends to first form a general framework (using global attention), then gradually refine local details (using local attention) driven by the global perspective (using global attention) shown in Figure 1. Inspired by the sketching process, we integrate the alternation process of local and global attention to EDT, and propose Attention Modulation Matrix (AMM) to modulate from the default global attention in self-attention mechanisms to local attention. AMM, functioning as a plug-in, which can be seamlessly integrated into diffusion transformers, enhancing image synthesis performance without necessitating additional training.

**Token relation-enhanced masking training strategy** The token compression in down-sampling modules may cause token information loss. Learning the relations among tokens can help token down-sampling modules compress tokens effectively. And it has been confirmed that masking training can enhance the DPMs' ability to learn relations among object parts in images [22]. We propose a novel masking training strategy to enhance the relation learning among tokens. Specifically, the full tokens are fed into EDT and the tokens masking is executed in down-sampling modules. This forces models to learn token relations before some of the tokens are masked. We compare our masking training method to the counterpart in MDT, both implemented on EDT. Our masking training method achieves better generation.

We summarize the contributions of our work: 1. We develop an Efficient Diffusion Transformer (EDT) framework and design a lightweight diffusion transformer architecture based on a comprehensive computation analysis. 2. Inspired by human sketching, we design EDT with an alternation process between global attention and location attention. Moreover, to the best of our knowledge, we introduce Attention Modulation Matrix for the first time, which improves the detail of generated images of

pre-trained diffusion transformers without any extra training cost. 3. We propose a novel token masking training strategy to enhance the token relation learning ability of EDT. 4. EDT has reached a new SOTA and achieves faster training and inference speed compared to existing representative works DiT and MDTv2. We conduct a series of exploratory experiments and ablation studies to analyze and summarize the key factors affecting the performance of EDT.

## 2 Method

### 2.1 Preliminaries

We briefly review several fundamental concepts necessary to understand classifier-free guidance class-condition diffusion models [11]. The primary objective of diffusion models is to learn a diffusion process that constructs a probability distribution for a specific dataset, subsequently enabling the sampling of new images. Given a classifier-free guidance class-condition diffusion model $\epsilon_\theta(x_t, c)$, the model can generate images of specific class $c$ from Gaussian noise over multiple denoising time steps. The model operates through two main processes: the forward and reverse processes. The forward process simulates training data $x_t$ to be denoised at time step $t$, by adding Gaussian noise $\epsilon_t \sim \mathcal{N}(0, \mathbf{I})$ to the original data $x_0$. This process is mathematically described by $q(x_t|x_0) = \mathcal{N}(x_t; \sqrt{\bar{\alpha}_t}x_0, (1 - \bar{\alpha}_t)\mathbf{I})$, where $\bar{\alpha}_t$ denotes a hyperparameter. The reverse process samples noise-reduced data $x_{t-1}$ based on noise data $x_t$ and class-condition $c$. The reverse process is represented as $p_\theta(x_{t-1} \mid x_t, c) = \mathcal{N}(x_{t-1}|\mu_\theta(x_t, c), \Sigma_\theta(x_t, c))$, where $\mu_\theta$ and $\Sigma_\theta$ are the statistics of $p_\theta$. By optimizing the variational lower-bound of the log-likelihood [26] $p_\theta(x_0)$ and reparameterizing $\mu_\theta$ as a noise prediction network $\epsilon_\theta$, the model can be trained using simple mean-squared error between the predicted noise $\epsilon_\theta(x_t, c)$ and the ground truth $\epsilon_t$ sampled Gaussian noise: $\mathcal{L}_\theta(x_t, c, \epsilon_t) = \|\epsilon_\theta(x_t, c) - \epsilon_t\|_2^2$. Additionally, $\epsilon_\theta(x_t, c)$ is a standard class-condition model; when $c = \emptyset$, it functions as an unconditional model. To allow the controllability of class-condition guidance, the prediction of models is further derived as $\hat{\epsilon}_\theta(x_t, c) = \epsilon_\theta(x_t, \emptyset) + \omega \cdot (\epsilon_\theta(x_t, c) - \epsilon_\theta(x_t, \emptyset))$, where $\omega \geq 1$ is class-condition guidance intensity.

In this work, we employ a classifier-free guidance class-condition diffusion transformer architecture operating on latent space. The pre-trained variational autoencoder (VAE) model [26] from LDM [11] remains frozen and is used to encode/decode the image/latent tokens.

### 2.2 Lightweight-design diffusion transformer

Transformer-based diffusion probabilistic models (DPMs) have demonstrated greater scalability and superior performance compared to CNN-based DPMs [20, 21, 22]. However, these models also entail significant computational overhead during both the training and inference phases. In response, we design a lightweight diffusion transformer architecture in this section. We undertake a computational complexity analysis of the transform-based diffusion model. Based on the empirical analysis of the number of tokens, token dimensions, FLOPs, and the number of parameters, we establish two design principles: (1) reducing the number of tokens to decrease the FLOPs in the self-attention module through the down-sampling module; (2) ensuring that the FLOPs of each EDT stage post a down-sampling module are significantly reduced compared to the stages prior to the down-sampling module, to effectively lower the overall FLOPs.

Building on the aforementioned design principles, we have redesigned and incorporated the down-sampling, up-sampling, and long skip connection modules into the transformer-based diffusion model, successfully achieving a reduction in FLOPs and increased inference speed. For instance, in comparison to DiT-S [21], our smaller version model EDT-S achieves an inference speed of 5.5 steps per second, versus 2.7 steps per second for DiT-S, effectively doubling the speed. Figure 2 illustrates the architecture of our lightweight-designed diffusion transformer. The model includes three EDT stages in the down-sampling phase, viewed as an encoding process where tokens are progressively compressed, and two EDT stages in the up-sampling phase, viewed as a decoding process where tokens are gradually reconstructed. These five EDT stages are interconnected through down-sampling, up-sampling, and long skip connection modules. Note that each EDT stage comprises several consecutive transformer blocks. For more details on the computational complexity analysis and model design, please refer to Appendix A.2. It is important to note that *the down-sampling and up-sampling phases can be viewed as encoding and decoding processes, respectively, aligning with*

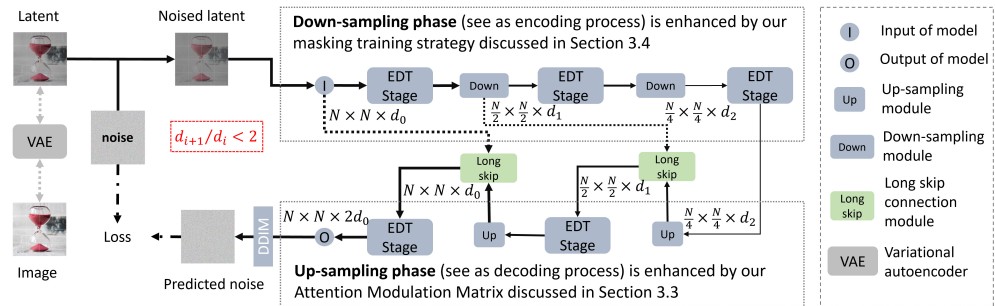

Figure 2: The architecture of lightweight-design diffusion transformer.

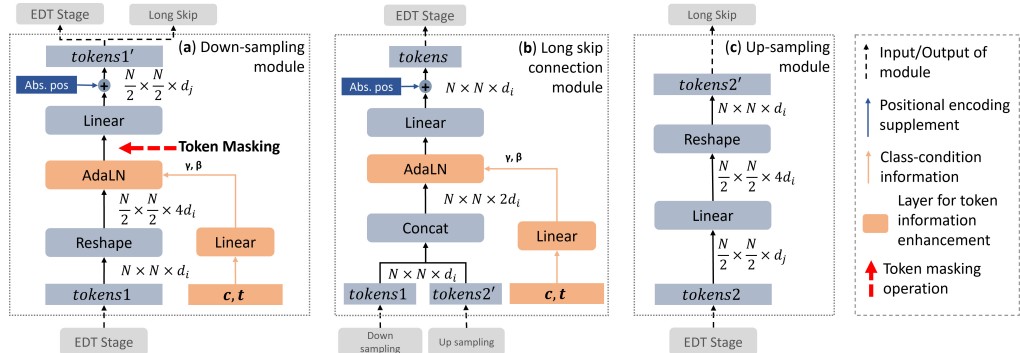

Figure 3: The design of down-sampling, long skip connection and up-sampling modules.

*the conceptualization of drawing pictures in human sketching.* These phases are crucial and will be further discussed in the following Section 2.3 and Section 2.4.

While we have successfully reduced the FLOPs of the model, the token merging operation in the down-sampling and long skip connection modules inevitably leads to a loss of token information, including the positional encoding and contextual data essential for class-condition generation of images. To mitigate this loss of token information, we propose two improvements, as illustrated in Figure 3: **token information enhancement** and **positional encoding supplement**.

**Token information enhancement** We enhance the contextual information required for class-condition generation by employing Adaptive Layer Normalization (AdaLN) before the token merging process. AdaLN adjusts the output by learning scaling factors $\gamma$ and bias coefficients $\beta$, which can scale, negate, or shut off the features [27]. By utilizing AdaLN, we modulate the tokens based on class conditions and time steps before merging, thereby preserving more contextual information and minimizing the loss of token information. As depicted in Figure 3, class-condition information is integrated by the $\gamma$ and $\beta$ of AdaLN in both the down-sampling and long skip connection modules.

**Positional encoding supplement** We restore the absolute positional encoding of tokens following the token merging process. As illustrated in Figure 3, after merging the tokens, we add the absolute positional encoding to the merged tokens at the end of both the down-sampling and long skip connection modules.

## 2.3 Making EDT "sketch" like a human

The lightweight design of EDT might compromise the quality of image synthesis. To enhance the detail fidelity in generated images, we have refined the decoding process (up-sampling phase) of EDT by imitating the process of human sketching. We begin by examining how attention shifts during the act of sketching by humans. Human cognition stores visual structures as a top-down hierarchy passing from general shape to the relationships between parts down to the detailed features of individual parts [23, 24]. This hierarchical structuring of visual information in the brain makes humans tend to follow a coarse-to-fine strategy in sketching [25]. As shown in Figure 1, the process of human

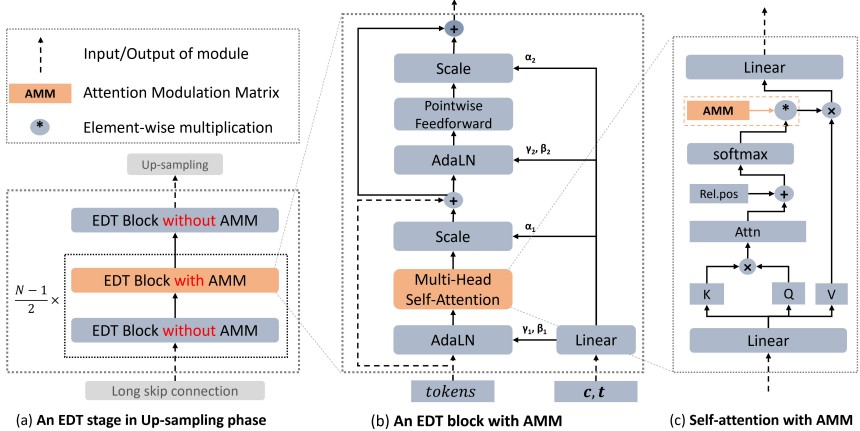

(a) **An EDT stage in Up-sampling phase**    (b) **An EDT block with AMM**    (c) **Self-attention with AMM**

Figure 4: The position of Attention Modulation Matrix (local attention) in an EDT stage in the up-sampling phase.

sketching tends to first form a general framework (using global attention), then gradually refine local details [24, 28] (using local attention) hinted by the global perspective (using global attention). Even when concentrating on a local detail, humans do not become completely detached from the overall framework. Therefore, humans periodically shift attention back to a global view to scrutinize the local detail and further fine-tune it [29, 30]. This process reflects the alternation of global and local attention in the human brain when sketching.

Inspired by the sketching process, we aim to integrate the alternation process of local attention and global attention to EDT. In the current series of diffusion transformers [20, 22, 21], only default global attention mechanisms are employed, which may lead to poor generation of local details. Therefore, we introduce the Attention Modulation Matrix (AMM) to enhance focus on local details. Moreover, to mimic the alternation process in the EDT, we alternately incorporate the AMM into the lightweight-design transformer diffusion architecture.

### 2.3.1 Integrating local attention into the up-sampling phase of EDT

To imitate the alternation between global and local attention like the act of humans drawing, we integrate local attention into the up-sampling phase of EDT by introducing Attention Modulation Matrix (AMM). In this section, we concentrate on imitating the alternation process of attention. A detailed discussion of AMM is deferred to Section 2.3.2. We align the decoding process of EDT with the humans drawing pictures. Consequently, we incorporate local attention (AMM) into the decoding process (up-sampling phase) of EDT. Figure 4 illustrates the placement of AMM (local attention) in an EDT stage. As depicted in Figure 4(a), we alternately configure EDT blocks with and without the AMM, thereby mimicking the alternation between global and local attention observed in drawing activities. The EDT block with AMM is shown in Figure 4(b). As shown in Figure 4(c), the AMM is integrated into the self-attention module. The AMM and the global attention score matrix are combined via a Hadamard product to modulate global attention into local attention.

### 2.3.2 Attention modulation matrix

We develop the Attention Modulation Matrix (AMM) to modulate the default global attention in self-attention mechanisms into local attention, which imitates the local attention of humans during the act of drawing. Humans typically concentrate on either the actively engaged parts or the most salient aspects of a visual scene [31]. When drawing a specific local region of an image, areas closer to the region of interest tend to exhibit stronger contextual relations and thus warrant increased attention. Conversely, areas further from the region of interest generally show weaker contextual relations and can be allocated less attention. Thus, we articulate the principle: *for a local region, the strength of attention on contextual relations within a specific region is inversely related to the distance between the local region and the specific region.* In the self-attention mechanism, we regard the attention score between tokens as an indicator of the strength of attention on contextual relations between regions. Similarly, we aim to modulate the strength of attention based on the distance among tokens

on the image. Based on this concept, we have developed the Attention Modulation Matrix (AMM), functioning as a plug-in, which can be seamlessly integrated into diffusion transformers, significantly enhancing image synthesis performance without necessitating additional training.

Formally, given a sequence of $N \times N$ tokens and its corresponding attention score matrix $\mathbf{A} \in \mathbf{R}^{N^2 \times N^2}$, we take two arbitrary tokens as an example to illustrate the formulation. The two arbitrary tokens are denoted as $Token_i, Token_r$ and their attention score $a_{ir}$, where $a_{ir} \in \mathbf{A}$. The coordinates of $Token_i$ and $Token_r$ correspond to the $(x_i, y_i)$ and $(x_r, y_r)$ in the original $N \times N$ tokens grid, where $i = Nx_i + y_i$ and $r = Nx_r + y_r$. The distance between these two tokens can be calculated by Euclidean distance $d_{ir} = \sqrt{(x_i - x_r)^2 + (y_i - y_r)^2}$, and we can derive the token distance matrix $\mathbf{D} \in \mathbf{R}^{N^2 \times N^2}$. We aim to modulate the global attention into local attention by multiplying the attention score matrix to the AMM, which is generated based on the token distance matrix $\mathbf{D}$. The modulation matrix generation function $F$ is designed with adherence to two principles: (1) the generation function should be monotonically decreasing within the interval $[0, d_{max}]$, ensuring that the modulation matrix elements are inversely correlated with distance, where $d_{max} = (N - 1)\sqrt{2}$ is the furthest distance, which is the distance between two diagonal opposite tokens; (2) the output range of this function should be limited to avoid significantly altering the original distribution of the attention score matrix. Based on the two principles, we utilize the monotonically decreasing interval in cosine function, $\cos(fd_{ir}), d_{ir} \in [0, d_{max}]$, where the monotonically decreasing interval can be flexibly adjusted by adjusting its period $T$ or frequency $f$. According to the $d_{max}$, we set $T = 4d_{max}$ and $f = \frac{2\pi}{T}$. Further, we employ $\cos(fd_{ir})$ as the exponent of the Euler's number $e$, thereby smoothing the values of the modulation matrix elements. We obtain the final modulation matrix generation function $F(d_{ir}) = ke^{\cos(fd_{ir})}$, which can flexibly scale the function value to $[k, ke]$ by scaling factor $k$. We empirically set $k = 0.5$ and the output range within $[\frac{1}{2}, \frac{e}{2}]$, which allows the modulation matrix elements to appropriately adjust the attention scores. In addition, we define an effective radius $R$ for local attention to exclude the interactions of tokens that occur over tokens with far distances. For each token pair, we only modulate the attention scores with distance $d_{ir} \leq R$, where $R$ is the effective radius for local attention. We set $R = \sqrt{(N-1)^2 + 4}$ based on experiments regarding the hyper-parameters of the AMM, detailed in Appendix A.3.3. And those $d_{ir} > R$, their attention scores are set to zero, indicating that tokens far from the region of interest exert less influence. Thus, we have the Attention Modulation Matrix $\mathbf{M} \in \mathbf{R}^{N^2 \times N^2}$, where $m_{ir} \in \mathbf{M}$ is defined as:

$$m_{ir} = \begin{cases} F(d_{ir}), & d_{ir} \leq R \\ 0, & d_{ir} > R \end{cases} \tag{1}$$

The modulated attention score element is $a'_{ir} = a_{ir} * m_{ir}$, where $a'_{ir} \in \mathbf{A}'$, and $\mathbf{A}'$ is the modulated attention score matrix. Further details about the entire process and an illustration of AMM can be found in Appendix A.3.1.

## 2.4 Token relation-enhanced masking training strategy

The ability to learn relations among object parts in images is crucial for image generation, as highlighted in MDT [22]. However, the down-sampling process in EDT inevitably leads to the loss of token information. Establishing relations among tokens can alleviate performance degradation caused by the loss of token information. To enhance the relation-learning ability in EDT, we introduce a relation-enhanced masking training strategy. Before detailing the proposed masking training strategy, we first explore the integration of MDT into EDT. Figure 5 (a) shows the masking training strategy of MDT. In MDT, the training loss $L$ contains two parts as shown in Eqn. 2.

$$L = L_{full} + L_{masked} = \mathcal{L}_\theta(x_t, c, \epsilon_t) + \mathcal{L}_\theta(mask * x_t, c, \epsilon_t) \tag{2}$$

$L_{full}$ is the loss when the input consists of the full token input, the $L_{masked}$ is the loss when the input consists of the remained tokens after masking, and $mask$ is a matrix to mask tokens randomly.

However, our analysis reveals that the masking training method used in MDT excessively focuses on masked region reconstruction at the expense of diffusion training, potentially leading to a degradation in image generation performance. Additionally, our evaluation of MDT is observed a conflict between the training objectives of $L_{full}$ and $L_{masked}$. Specifically, as $L_{full}$ decreases, $L_{masked}$ increases, and vice versa, demonstrating the conflicting nature of these training objectives. To mitigate this conflict and allow the model to focus on the diffusion generation task, as shown in Figure 5 (b), we

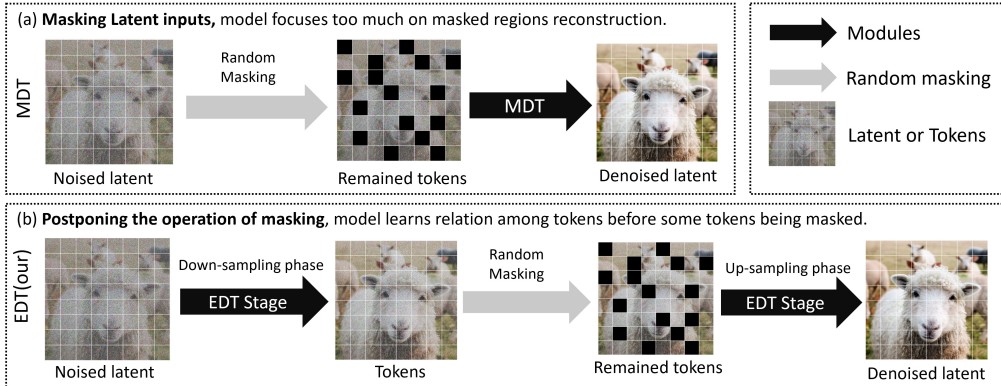

Figure 5: Token relation-enhanced masking training strategy. MDT is fed the remained tokens after token masking into models. EDT is fed full tokens into shallow EDT blocks, and the operation of token masking is performed in down-sampling modules.

design the token relation-enhanced masking training strategy, which feeds full tokens into shallow blocks and postpones the token masking operation to occur within the down-sampling modules. This strategy is designed to facilitate learning relationships among tokens and reduce the loss of token information without the issues arising from conflicting training objectives. When training, the masked tokens are unseen to the EDT blocks following the operation of masking, which forces the EDT blocks before the operation of masking to learn the relations among tokens. As the relations among tokens are learned, the key information in each token is dispersed and stored across various tokens. This avoids reliance on certain tokens and reduces the loss of token information from compression in down-sampling. Figure 3(a) shows, with a red arrow in the down-sampling module, the specific point at which token masking is performed. After passing through the down-sampling modules, the EDT stages in up-sampling phase generate images solely relying on the remained tokens. The loss function of EDT with token relation-enhanced masking training strategy is shown in Eqn. 3, where the token masking operation is executed in the down-sampling modules.

$$L = L_{full} + L_{masked} = \mathcal{L}_\theta(x_t, c, \epsilon_t) + \mathcal{L}_\theta(x_t, c, mask, \epsilon_t) \tag{3}$$

We discover that EDT is particularly well-suited for masking training due to its up-sampling modules, which inherently are used for the reconstruction of tokens. Unlike MDT, which requires an additional interpolator module for reconstructing masked tokens, EDT eliminates the need for such a module, thereby reducing unnecessary training overhead associated with an interpolator compared to MDT. For a more detailed analysis, please refer to Appendix A.4.1.

## 3 Experiment

### 3.1 Implementation Details

**Models**: We develop three different sizes of EDT including small (EDT-S), base (EDT-B) and extra large (EDT-XL), each using a patch size of two. Details regarding token dimensions, head numbers, and parameter counts are provided in Table 5 of Appendix A.2.2. **Training and evaluation**: The training dataset is ImageNet [32] with 256×256 and 512×512 resolution. For a fair comparison, we follow the training settings of MDTv2 [33]. EDT uses the Adan [34] optimizer with a global batch size of 256 and without weight decay. The learning rate linearly decreases from 1e-3 to 5e-5 over 400k iterations. **Masking training strategy**: We set the mask ratio $0.4 \sim 0.5$ in the first down-sampling module, and $0.1 \sim 0.2$ in the second. The investigation of mask ratio refers to Appendix A.4.2. **GPUs**: Training is conducted on eight L40 48GB GPUs, while the speed test for inference is performed on a single L40 48GB GPU. **Evaluation metrics**: Common metrics such as Fre'chet Inception Distance (FID) [35], sFID [36], Inception Score (IS) [37], Precision, and Recall [38] are used to assess the model performance. The training speed is evaluated by iterations per second, and inference speed is assessed by steps per second using a batch size of 256 in FP32. For fair comparison, we follow [21, 33] and employ the TensorFlow evaluation suite from ADM [4], reporting FID-50K results with 250 DDIM [10] sampling steps. These metrics are reported by default without the classifier-free guidance.

Table 1: The comparison with existing SOTA methods on class-conditional image generation without classifier-free guidance on ImageNet 256×256. We report the training speed (T-speed), inference speed (I-speed), and memory consumption (Mem.) of inference. The EDT* denotes the EDT without our proposed token relation-enhanced masking training strategy.

| Model | Cost↓ (Iter×BS) | Params. (M) | T-speed (iter/s) | GFLOPs↓ | I-Speed (step/s) | Mem. (MB) | FID↓ |
|---|---|---|---|---|---|---|---|
| DiT-S[21] | 400K×256 | 32.90 | 12.50 | 6.06 | 2.70 | 4296 | 68.40 |
| SD-DiT-S[39] | 400K×256 | 32.90 | - | - | - | - | 48.39 |
| **EDT-S*(our)** | 400K×256 | 38.30 | **13.20** | **2.66** | **5.50** | 4268 | 38.73 |
| MDTv2-S[33] | 400K×256 | 33.10 | 2.25 | 6.07 | 2.40 | 4902 | 39.50 |
| **EDT-S (our)** | 400K×256 | 38.30 | 8.86 | **2.66** | **5.50** | 4268 | **34.27** |
| DiT-B[21] | 400K×256 | 130.30 | 4.30 | 23.01 | 1.11 | 8978 | 43.47 |
| SD-DiT-B[39] | 400K×256 | 130.30 | - | - | - | - | 28.62 |
| **EDT-B*(our)** | 400K×256 | 152.00 | **5.80** | **10.20** | **2.20** | 8584 | 23.19 |
| MDTv2-B[33] | 400K×256 | 130.80 | 1.42 | 23.02 | 0.96 | 9212 | 19.55 |
| MDTv2-B[33] | 1600K×256 | 130.80 | 1.42 | 23.02 | 0.96 | 9212 | 13.60 |
| EDT-B(our) | 400K×256 | 152.00 | 4.03 | 10.20 | 2.20 | 8584 | 19.18 |
| **EDT-B(our)** | 1000K×256 | 152.00 | 4.03 | **10.20** | **2.20** | 8584 | **13.58** |
| ADM[4] | 1980k×256 | 554.00 | - | 1120.00 | - | - | 10.94 |
| LDM-4[11] | 178k×1200 | 400.00 | - | 104.00 | - | - | 10.56 |
| DiT-XL[21] | 400K×256 | 674.80 | 0.93 | 118.64 | 0.25 | 17538 | 19.47 |
| SD-DiT-XL[39] | 1300K×256 | 740.60 | - | - | - | - | 9.01 |
| **EDT-XL*(our)** | 400K×256 | 698.40 | **1.49** | **51.83** | **0.51** | **14486** | 10.48 |
| MDTv2-XL[33] | 400K×256 | 675.80 | 0.51 | 118.69 | 0.23 | 23436 | 7.70 |
| **EDT-XL(our)** | 400K×256 | 698.40 | 0.98 | **51.83** | **0.51** | **14486** | **7.52** |

Table 2: The comparison with existing transformer-based models on class-conditional image generation without classifier-free guidance on ImageNet 512×512.

| Model | T-speed (iter/s) | GFLOPs↓ | FID↓ | IS↑ | sFID↓ |
|---|---|---|---|---|---|
| DiT-S | **2.26** | 31.42 | 85.21 | 23.68 | 13.53 |
| MDTv2-S | 0.53 | 31.46 | **51.16** | 29.94 | 8.57 |
| EDT-S(our) | 1.63 | **13.25** | 51.84 | 29.92 | **7.86** |

## 3.2 Comparison with SOTA transformer-based diffusion methods

To validate the enhancements in speed and generation performance of EDT, we conducted comparisons with both classical methods [4, 11, 21] and recent advancements [33, 39, 40].

**Experiment on ImageNet 256×256** The result of image generation without classifier-free guidance is shown in Table 1. Our comparisons across three different sizes demonstrate that EDT consistently achieves the best FID scores: EDT-S scored an FID of 34.2, EDT-B scored 19.1, and EDT-XL scored 7.5. Notably, EDT also showed significant reductions in GFLOPs compared to the second-best MDTv2 across all sizes (2.66 GFLOPs vs. 6.07 GFLOPs, 10.2 GFLOPs vs. 23.02 GFLOPs, 51.83 GFLOPs vs. 118.69 GFLOPs). Moreover, EDT exhibited the lowest memory consumption during inference across all three sizes, underscoring the efficiency of our lightweight design. We further investigated the training speed of EDT. Given that both EDT and MDTv2 incorporate additional training strategies, we specifically compared the training speeds of these two models. Additionally, we assessed the training speed of EDT without the masking training strategy (denoted as EDT*) against other methods. In both scenarios, EDT trained faster than the baseline models. For example, EDT-XL* achieved a training speed of 1.49 iter/s, compared to 0.93 iter/s for DiT-XL. In comparison to MDTv2-XL, which trained at 0.51 iter/s, EDT-XL was nearly twice as fast at 0.98 iter/s. We further perform the experiment on **the image generation with classifier-free guidance**. The result is shown in Table 13 of Appendix A.5.1. Under the same training cost, EDT-S-G achieves the lowest FID score compared to MDTv2-S-G (9.89 vs. 15.62). Overall, these findings confirm that EDT significantly enhances both the speed and performance of image synthesis. Additionally, the training cost of EDT is efficient. We include a training cost analysis of EDT in Appendix A.2.3.

**Experiment on ImageNet 512×512** As shown in Table 2, we train DiT-S, MDTv2-S and EDT-S on ImageNet 512×512 for 60 epochs. Under the same training cost, both MDTv2-S and EDT-S achieve better FID scores than DiT-S (51.16 vs. 85.21, 51.84 vs. 85.21). In terms of training speed and inference overhead, EDT-S is 3.07 and 2.37 times faster than MDTv2-S respectively. This indicates that EDT achieves competitive image generation performance with lower resource overhead.

Table 3: Results on various models with (w) AMM and without (w/o) AMM. These models are trained for 400k iterations by default. We evaluate models using FID scores.

| Model | W/o AMM | W AMM | Model | W/o AMM | W AMM |
|-------|---------|-------|-------|---------|-------|
| EDT-S* | 50.90 | **38.73** | EDT-S | 46.90 | **34.27** |
| EDT-B* | 33.19 | **23.19** | EDT-B | 26.30 | **19.18** |
| EDT-XL* | 14.92 | **10.48** | EDT-XL | 12.80 | **7.52** |
| DiT-S | 67.16 | **63.11** | MDTv2-S | 39.02 | **31.89** |
| DiT-XL | 18.48 | **14.73** | DiT-XL-7000k | 9.62 | **3.75** |

### 3.3 Ablation Study

#### 3.3.1 Attention Modulation Matrix

Generated images using EDT-XL **without AMM**  Generated images using EDT-XL **with AMM**

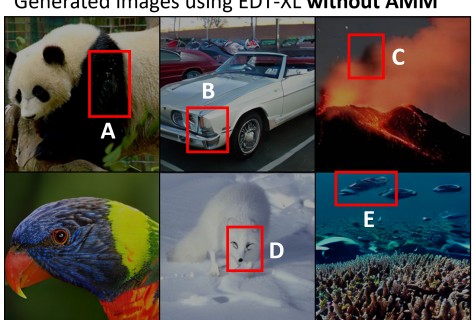 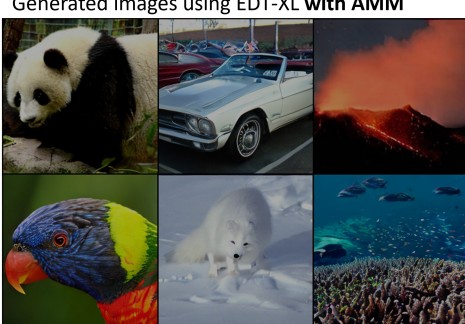

Figure 6: EDT-XL with AMM achieves more realistic visual effects. **Area A:** There are some blue stains on the panda's arm. **Area B:** An unreasonable gray area. **Area C:** Black smoke in the red fog. **Area D:** Unrealistic eyes of the fox. **Area E:** Fish with an odd shape. The parrot image generated by EDT-XL without AMM is realistic. And the parrot image generated by EDT-XL with AMM remains equally realistic. The add of AMM does not negatively affect the original quality.

**Quantitative analysis** We demonstrate the effectiveness and broad applicability of AMM across various models by comparing the FID scores between models with AMM and without AMM in Table 3. Extensive results show that the pre-trained models with AMM consistently outperform models without AMM, thereby verifying the generality and effectiveness of AMM. For instance, MDTv2-S with AMM achieves a better FID score than MDTv2-S without AMM (31.89 vs. 39.02). Using AMM enhances the FID of DiT-XL from 18.48 to 14.73. EDT-XL also has a lower FID score of 7.52 compared to EDT-XL without AMM of an FID score of 12.8.

**Qualitative analysis** We demonstrate the effectiveness of AMM by comparing the synthesis images from EDT-XL and DiT-XL with and without the AMM. As shown in Figure 6, the red boxes highlight the unrealistic regions in the images generated by EDT-XL without AMM. In the corresponding regions of the images generated by EDT-XL with AMM, the results appear more realistic. Moreover, the parrot image generated by EDT-XL without AMM is realistic and the parrot image generated by EDT-XL with AMM still remains equally realistic. This visual analysis demonstrates the effectiveness of the AMM plugin. Please refer to A.3, and A.5.2 for more analysis about AMM. While AMM is effective, there is potential for improvement. Please refer to A.6 for details regarding its limitations.

#### 3.3.2 Lightweight-design diffusion transformer

We investigate the effectiveness of the key components in our proposed diffusion transformer architecture. We denote the token information enhancement as TIE and the positional encoding supplement

Table 4: The ablation study of the key components of the lightweight-design and masking training strategy of EDT. The experiment is conducted on the small-size EDT model (W/o AMM).

| Model | TIE | PES | masking training strategy of MDT | masking training strategy of EDT | FID↓ | IS↑ |
|---|---|---|---|---|---|---|
| Baseline | ✗ | ✗ | ✗ | ✗ | 53.90 | 29.29 |
| A | ✗ | ✓ | ✗ | ✗ | 52.76 | 29.38 |
| B | ✓ | ✗ | ✗ | ✗ | 52.13 | 30.60 |
| C | ✓ | ✓ | ✗ | ✗ | **50.90** | **31.02** |
| D | ✓ | ✓ | ✓ | ✗ | 49.60 | 33.11 |
| E | ✓ | ✓ | ✗ | ✓ | **46.90** | **35.40** |

as PES in Table 4. Model A incorporates TIE without PES, with an FID score of 52.8. Model B integrates PES without TIE, with an FID score of 52.1. Model C represents EDT-S*, and utilizes both components, with an FID score of 50.9. Upon comparing Models A and C, we observed that the usage of token information enhancement leads to an improved FID score, decreasing from 52.8 to 50.9. Similarly, the comparison between Models B and C demonstrates that the addition of a positional encoding supplement also results in a better FID score, reducing from 52.1 to 50.9. The experimental results confirm the effectiveness of both components in enhancing model performance.

### 3.3.3 Token relation-enhanced masking training strategy

We investigate the effectiveness of the token relation-enhanced masking training strategy and compare it with the training strategy used in MDT in Table 4. Model C does not employ any masking training strategy, with an FID score of 50.9 and an IS score of 31.0. Model D, a small-size EDT trained using the masking strategy of MDT, with an FID score of 49.6 and an IS score of 33.1. Model D shows only a slight improvement compared to Model C. Model E is a small-size EDT trained with the masking training strategy of EDT, with an FID score of 46.9 and an IS score of 35.4. Model E achieved the best performance in terms of both FID and IS. This result suggests that the masking training strategy of EDT successfully improves performance by enhancing the learning ability of token relations.

## 4 Conclusions

In this work, we propose the Efficient Diffusion Transformer (EDT) framework, which includes a lightweight-design of diffusion transformer, a training-free Attention Modulation Matrix (AMM) inspired by human-like sketching, and the token relation-enhanced masking training strategy. Our lightweight-design reduces the number of tokens through down-sampling to lower computational costs. We redesigned down-sampling module and masking training strategy to address token information loss caused by the reduction of tokens. During inference, we introduce local attention through AMM, further enhancing image generation performance. Extensive experiments demonstrate that the EDT surpasses existing SOTA methods in both inference speed and image synthesis performance.

## 5 Related Work

**Diffusion Probabilistic Models**: Denoising diffusion probabilistic models (DDPM) [1], have marked a significant advancement in generative models. DDPM improves image generation by progressively reducing noise. ADM [4] innovates further by introducing a classifier-guided approach to refine the balance between image diversity and fidelity. Subsequent developments include a classifier-free method [9], which increases the flexibility of diffusion models by eliminating classifier constraints. DiT [21] replacing U-Net with transformer in LDM [11], achieving superior scalability. However, transformers-based models are computationally intensive. **Efficient Diffusion**: Various methods have been developed to enhance the efficiency of diffusion models. DDIM [10] redefines the diffusion process as non-Markovian, speeding up-sampling by removing dependencies on sequential time steps in DDPM [1]. LDM [11] reduces computational demands by transforming high-resolution images into a latent space for diffusion, thus balancing complexity with image detail. Current research in lightweight diffusion transformers is limited but offers potential for further efficiency improvements in diffusion model technologies. For more related work, please refer to Appendix A.1.

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

# A Appendix

The appendix is structured as follows. In Appendix A.2, we provide a computational complexity analysis of DiT and the lightweight architecture design of EDT. In Appendix A.3, we show the computational process of the AMM and explore the usage methods of AMM, hyper-parameters. In Appendix A.4, we analyze the loss changes of different masking training strategies and explore the selection of mask ratio. In Appendix A.5, we report additional experimental results and visualization of generated images by EDT-XL-2000k. Appendix A.6 discusses the limitations.

## A.1 Related Work

**Diffusion Probabilistic Models** In the field of generative models, diffusion models have achieved significant success. The Denoising diffusion probabilistic model (DDPM) [1] is the most classic denoising diffusion model, which generates new images by gradually removing noise from the noising data. NCSN [2] guides the data from low to high probability density areas by predicting the gradient of the data distribution. SDE [3] considers the noise addition and removal process as continuous and uses stochastic differential equations to guide the data generation process, theoretically unifying DDPM and NCSN and providing a new perspective for generative models. ADM [4] introduces a classifier-guided mode, using classifier gradients to guide the diffusion model's generation process, balancing the diversity and fidelity of generated images. Building on ADM, [9] proposes a classifier-free guidance method, removing the constraints of the classifier and significantly enhancing the flexibility of conditional diffusion generation. Compared to earlier works based on the U-Net backbone, models based on the Transformers backbone achieved better results. U-ViT [20] adds U-Net's skip connections to ViT [41] for diffusion generation in the pixel space, showcasing the broad prospects for diffusion transformers. DiT [21] replaces the U-Net backbone with ViT based on LDM [11], surpassing previous U-Net-based performances and showing good scalability. However, due to the intensive computational nature of transformers, integrating them effectively into diffusion models still presents significant challenges.

**Efficient Diffusion** Several methods have been proposed previously to improve the inference efficiency of diffusion models. DDIM [10] defines the diffusion process as a non-Markov process, thus eliminating the dependency on adjacent time steps in the reverse process of DDPM[1], achieving faster sampling speeds. LDM [11] compresses images from high-resolution pixel space to latent space and performs diffusion generation in latent space, achieving a balance between computational complexity and image detail. [42] introduces a method of gradual distillation for unconditional and classifier-guided diffusion models, which optimizes higher iteration counts into lower ones. Building on the work of [42], [43] proposes a two-stage distillation method for classifier-free guided diffusion models. Studies [22, 44] improves the training speed of DiT [21] by masking inputs. HDiT [40] introduces an hourglass-structured diffusion transformer for generating high-resolution images, which requires 70% fewer inference FLOPs in pixel space compared to DiT [21], but since HDiT operates in pixel space with high input and output resolution, it still demands substantial computational resources. ToddlerDiffusion [45] proposes a novel approach that extends the diffusion framework into modality space, decomposing the complex task of RGB image generation into simpler, interpretable stages. Currently, research on lightweight diffusion transformers is relatively sparse, but it is orthogonal to the methods in [10, 42, 43]. Building on existing research, lightweight diffusion transformers will further enhance the efficiency of diffusion models, which is one of the core contributions of this paper.

## A.2 Computational analysis and lightweight design

We preset two lightweight design rules: (1) to reduce the FLOPs in the self-attention module, we decrease the number of tokens by token down-sampling; (2) to guarantee the total FLOPs significant reduction, the FLOPs of the EDT blocks after down-sampling should be significantly reduced compared to the EDT blocks before down-sampling. Utilizing the down-sampling module is key to achieving these two design rules.

We first analyze the applicable scenarios of conventional down-sampling modules in Appendix A.2.1. In Appendix A.2.2, we improve the down-sampling module to adapt to our scenario of latent diffusion models. Then, in Appendix A.2.3, the training costs of EDT, DiT, and MDT are reported.

Table 5: The model details of EDT across three different sizes.

| Model | Params. | Number of blocks | Blocks in each stage | Dimensions in each stage | Heads in each stage |
|-------|---------|------------------|---------------------|--------------------------|---------------------|
| EDT-S/2 | 32.2M+6.1M | 12 | [2,2,2,3,3] | [312,416, 520,416,312] | [6,8, 10,8,6] |
| EDT-B/2 | 128M+24.1M | 12 | [2,2,2,3,3] | [624,832, 1040,832,624] | [12,16, 20,16,12] |
| EDT-XL/2 | 644M+54.4M | 28 | [6,4,4,7,7] | [936,1248, 1560,1248,936] | [18,24, 30,24,18] |

### A.2.1 Applicable scenarios of the conventional down-sampling module

Firstly, we review the conventional down-sampling module. We have a token sequence of shape $N \times N$, where $n$ is the number of tokens ($n = N^2$). The dimension of token is denoted by $d$. The number of heads in multi-head attention is $h$, and $D$ denotes the dimension of a head. **Conventional down-sampling module [46, 47, 48] reduces the number of tokens by a factor of $k^2$ and increase the token dimensions by a factor of $k$ at the same time, where $k$ is down-sampling factor ($k \geq 2$).** Large down-sampling factor $k$ will cause too many tokens to be merged, harming performance. Therefore, $k$ is generally equal to 2. That means, in a $N \times N$ token sequence, we reduce the number of tokens $n$ by down-sampling adjacent 2 tokens into one. Then we obtain a $\frac{N}{2} \times \frac{N}{2}$ token sequence with fewer tokens and the token dimensions increase to $2d$ from $d$. Table 6 shows FLOPs analysis of a DiT block. The total FLOPs $F$ is $2n^2d + 12nd^2 + 6d^2$ and the number of parameters $P$ is $18d^2$ in a DiT block.

Now we analyze how much can down-sampling module reduce FLOPs , and its applicable scenarios. To facilitate derivation and calculation, we set $j = \frac{n}{d}$, where $j$ is the proportional coefficient between the number of tokens and the dimension of token. We explore the relationship between proportional coefficient $j$ and FLOPs drop ratio $\rho$ after using conventional down-sampling.

In Figure 7, before down-sampling, the total FLOPs of a DiT block $F$ is $2j^2d^3 + 12jd^3 + 6d^2$, where $n = N^2$ and $j = \frac{n}{d}$. After feeding tokens into conventional down-sampling, the number of tokens $n'$ is $\frac{n}{4}$ and token dimensions $d'$ is $2d$. Then feeding these tokens into a DiT block, the total FLOPs of the DiT block becomes $F' = 2n'^2d' + 12n'd'^2 + 6d'^2 = \frac{n^2d}{4} + 12nd^2 + 24d^2 = \frac{j^2d^3}{4} + 12jd^3 + 24d^2$ and the number of parameters in the DiT block is $P' = 72d^2$.

Comparing DiT blocks after and before the down-sampling module, the FLOPs drop is $\bigtriangledown F = F - F' = \frac{7j^2d^3}{4} - 18d^2$. We calculate the relationship between the FLOPs drop ratio $\rho = \frac{\bigtriangledown F}{F}$ and the proportional coefficient $j$ :

$$
\begin{aligned}
\rho &= \frac{\bigtriangledown F}{F} \\
&= \frac{\frac{7j^2d^3}{4} - 18d^2}{2j^2d^3 + 12jd^3 + 6d^2} \\
&= \frac{7j^2d - 72}{8j^2d + 48jd + 24} \\
&< \frac{7j^2d}{8j^2d + 48jd} \\
&= \frac{7j}{8j + 48} \\
&= \frac{7}{8 + \frac{48}{j}}
\end{aligned}
\tag{4}
$$

The Eqn.4 shows that $\frac{7j}{8j+48}$ is the upper limit of the FLOPs drop ratio $\rho$ and proportional to $j$. **Significant reductions in FLOPs can be achieved through down-sampling, only when the number of tokens $n$ is larger than the token dimensions $d$, namely $j > 1$.** For instance, when

Table 6: FLOPs in a DiT block

| Module | Operator | Input Shape | Params | Output Shape | FLOPs |
|---|---|---|---|---|---|
| AdaLN | fc | $1 \times d$ | $d \times 6d$ | $1 \times 6d$ | $6d^2$ |
| Attention | Att-kqv | $1 \times n \times d$ | $d \times 3d$ | $1 \times n \times 3d$ | $3nd^2$ |
| | K@Q | K: $1 \times h \times n \times D$ 
 Q: $1 \times h \times D \times n$ | - | $1 \times h \times n \times n$ 
 $(d = hD)$ | $dn^2$ |
| | Att@V | Att: $1 \times h \times n \times n$ 
 V: $1 \times h \times n \times D$ | - | $1 \times h \times n \times D$ 
 $(d = hD)$ | $dn^2$ |
| | fc | $1 \times n \times d$ | $d \times d$ | $1 \times n \times d$ | $nd^2$ |
| FFN | fc1 | $1 \times n \times d$ | $d \times 4d$ | $1 \times n \times 4d$ | $4nd^2$ |
| | fc2 | $1 \times n \times 4d$ | $4d \times d$ | $1 \times n \times d$ | $4nd^2$ |
| Total | | $1 \times n \times d$ | $18d^2$ | $1 \times n \times d$ | $2n^2d + 12nd^2 + 6d^2$ |

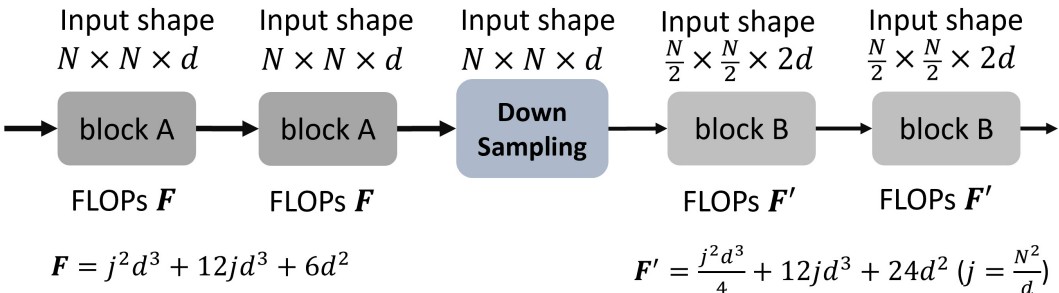

Figure 7: Input shape and FLOPs of DiT block before and after the conventional down-sampling module.

$j = 6$ (or $n = 6d$), the FLOPs drop ratio $\rho$ is about 50%; and when $j = 1$ (or $n = d$), the FLOPs drop ratio $\rho$ is about 12.5%.

However, in our application scenarios of latent diffusion transformer, where $\frac{n}{d} = j < 1$ , the FLOPs drop ratio $\rho$ is smaller than 12.5%. Only using conventional down-sampling in our scenarios of latent diffusion transformers can hardly reduce the computational complexity of the DiT blocks. So it does not meet the design rule (2): the FLOPs of the blocks after down-sampling should be significantly reduced compared to the blocks before down-sampling, to guarantee the total FLOPs reduction.

### A.2.2 Redesign the down-sampling module

Now, we redesign the down-sampling to make the architecture meet rule (2).

According to Appendix A.2.1, before the down-sampling module, the total FLOPs of a DiT block is $F = 2n^2d + 12nd^2 + 6d^2 = 2j^2d^3 + 12jd^3 + 6d^2$. And after the down-sampling module, the total FLOPs of a DiT block after down-sampling is $F' = \frac{n^2d}{4} + 12nd^2 + 24d^2$. Among the three items of $F$ and $F'$, the second term $12nd^2$ predominates due to $d > n$. However, this term isn't affected by the down-sampling process. This results from the conventional down-sampling module, which reduces the number of tokens by a factor of $k^2$ and increases the token dimensions by a factor of $k$ at the same time, namely $12nd^2 = 12 \times \frac{n}{k^2} \times (kd)^2$

To reduce the second item $12nd^2$, we should redesign the process of down-sampling module: we reduce the number of tokens by down-sampling factor $k = 2$ and increase the token dimensions by a factor of $r$ at the same time, where $r$ is token dimension expansion coefficient and $1 < r < k$. This design makes the second item reduce, namely $12 \times \frac{n}{k^2} \times (rd)^2 < 12nd^2$, and the total FLOPs $F' = \frac{rn^2d}{8} + 3nr^2d^2 + 6r^2d^2 = \frac{rj^2d^3}{8} + 3jr^2d^3 + 6r^2d^2$ in the blocks after our down-sampling

module. The FLOPs drop ratio $\rho$ is:

$$
\begin{aligned}
\rho &= \frac{\triangledown F}{F} = \frac{F - F'}{F} = 1 - \frac{F'}{F} \\
&= 1 - \frac{\frac{rn^2d}{8} + 3nr^2d^2 + 6r^2d^2}{2n^2d + 12nd^2 + 6d^2} \\
&= 1 - \frac{rn^2 + 24ndr^2 + 48dr^2}{16n^2 + 96nd + 48d} \\
&= 1 - \frac{rj + 24r^2 + \frac{48r^2}{n}}{16j + 96 + \frac{48}{n}} \\
&\approx 1 - \frac{rj + 24r^2}{16j + 96} \\
&= r\left(\frac{1}{r} - \frac{j + 24r}{16j + 96}\right) \\
&> r\left(\frac{1}{r} - \frac{j + 48}{16j + 96}\right) \\
&= r\left(\frac{1}{r} - \frac{1}{16 - \frac{672}{j+48}}\right) \\
&\geq r\left(\frac{1}{r} - \frac{1}{16 - \frac{672}{1+48}}\right) \\
&= 1 - 0.43r
\end{aligned}
\tag{5}
$$

In Eqn.5, $\rho = 1 - \frac{rj+24r^2}{16j+96}$ shows that the FLOPs drop ratio $\rho$ is inversely proportional to the token dimension expansion coefficient $r$ and $1 - 0.43r$ is the lower bound of $\rho$. We can adjust the FLOPs of the DiT block after down-sampling by adjusting the token dimension expansion coefficient $r$. When $r$ is within the range $[1, k]$ and $k = 2$, the FLOPs drop ratio $\rho$ falls within the interval $[0.57, 0.14]$. To make the number of parameters in EDT network approximate to that in other works (DiT and MDT), we set $r \approx 1.25$. Namely, after a down-sampling, we reduce the number of tokens by a factor of 4 and increase the token dimensions by a factor of $1.25$ at the same time. This leads to about 47% FLOPs drop ratio of the block after down-sampling compared to that before down-sampling, which meets the requirement of rule (2).

Table 5 shows three different sizes of EDT. The number of blocks is consistent with that in the DiT network of the corresponding size. The number of parameters in the EDT network is also approximate to that in DiT. Our parameters consist of two parts, one is the EDT blocks parameter, and the other is the sampling module and long-skip connection module. In the table, the number of parameters is written as the sum of these two parts.

Figure 2 illustrates the architecture of our lightweight-designed diffusion transformer. The model includes three EDT stages in the down-sampling phase, viewed as an encoding process where tokens are progressively compressed, and two EDT Stages in the up-sampling phase, viewed as a decoding process where tokens are gradually reconstructed. These five EDT stages are interconnected through down-sampling, up-sampling, and long skip connection modules. The 'blocks in each stage' in Table 5 displays how many blocks there are in the corresponding EDT stage.

### A.2.3 Training costs

On ImageNet, we estimated the training cost of EDT, MDTv2, and DiT on a 48GB-L40 GPU in Table 7, using a batch size of 256 and FP32 precision. EDT achieves the best performance with a low training cost. GPU days refer to the days required for training the models on a single L40 GPU.

### A.3 Exploring Attention Modulation Matrix

During the sketching process, humans alternately use global attention and local attention. Therefore, we design Attention Modulation Matrix (AMM), to introduce local attention into the self-attention

Table 7: Training cost of EDT, MDTv2, and DiT on ImageNet

| Model | Resolution | Epochs | Cost (Images) | GPU days | GFLOPs↓ | FID |
|---|---|---|---|---|---|---|
| EDT-S* | 256×256 | 80 | 102M | 2.75 | 2.66 | 38.73 |
| DiT-S | 256×256 | 80 | 102M | 2.96 | 6.06 | 68.40 |
| MDTv2-S | 256×256 | 80 | 102M | 16.47 | 6.07 | 39.50 |
| EDT-B | 256×256 | 80 | 102M | 9.19 | 10.20 | 19.18 |
| DiT-B | 256×256 | 80 | 102M | 8.62 | 23.01 | 43.47 |
| MDTv2-B | 256×256 | 80 | 102M | 26.17 | 23.02 | 19.55 |
| EDT-XL | 256×256 | 80 | 102M | 37.79 | 51.83 | 7.52 |
| DiT-XL | 256×256 | 80 | 102M | 39.82 | 118.64 | 19.47 |
| MDTv2-XL | 256×256 | 80 | 102M | 72.62 | 118.69 | 7.70 |
| EDT-S | 512×512 | 60 | 77M | 19.02 | 13.25 | 51.84 |
| DiT-S | 512×512 | 60 | 77M | 12.26 | 31.42 | 85.21 |
| MDTv2-S | 512×512 | 60 | 77M | 51.96 | 31.46 | 51.16 |

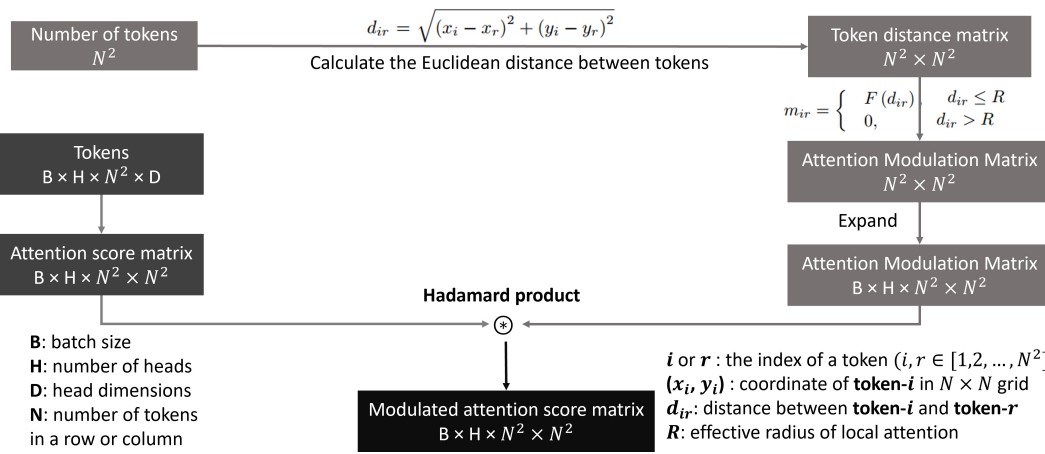

Figure 8: The process of modulating the attention score matrix and the changes in tensor shape.

module which uses global attention by default. Appendix A.3.1 shows the process of modulating the attention score matrix and the changes in tensor shape. In Appendix A.3.2, we explore the usage and arrangement of AMM in models. In Appendix A.3.3, we explore the settings of the hyper-parameters of AMM. The computational cost of AMM is discussed in Appendix A.3.4.

### A.3.1 The process of modulating the attention

The process of modulating the attention score matrix and the changes in tensor shape are shown in Figure 8. Image can be split into $N^2$ patches and each token is the feature of a patch. Each token (patch) corresponds to a rectangular area of the image and has a corresponding 2-D coordinate $(x, y)$ in the image grid. We calculate an Euclidean distance value $d$ for each pair of tokens, resulting in a distance matrix $\mathbf{D}$, which is an $N^2 \times N^2$ tensor. Based on the distance matrix, we generate modulation values $m$ via the modulation matrix generation function $F(d)$, which assigns lower modulation values to tokens that are farther apart. These modulation values form an Attention Modulation Matrix (AMM), another $N^2 \times N^2$ tensor. **Importantly, we integrate the AMM into the pre-trained EDT without any additional training.** The attention modulation matrix is calculated when the model is instantiated. During inference, the modulated attention score matrix is obtained by performing a Hadamard product between the attention modulation matrix and the attention score matrix.

Table 8: Comparison of adding AMM into EDT-S* during training versus inference on ImageNet $256 \times 256$.

| Model | adding AMM when training | adding AMM when inference | FID↓ | IS↑ |
|---|---|---|---|---|
| A | ✗ | ✗ | 50.9 | 31.0 |
| B | ✓ | ✓ | 52.1 | 30.3 |
| C | ✗ | ✓ | **38.7** | **36.4** |

Table 9: Performance of EDT-S* with varying insertion points of AMM on ImageNet $256 \times 256$.

| Model | AMM in encoder | AMM in decoder | Alternately inserting | FID↓ | IS↑ |
|---|---|---|---|---|---|
| A | ✗ | ✗ | ✗ | 50.9 | 31.0 |
| B | ✓ | ✓ | ✓ | 60.4 | 24.9 |
| C | ✗ | ✓ | ✗ | 44.4 | 35.1 |
| D | ✗ | ✓ | ✓ | **38.7** | **36.4** |

### A.3.2   When and where to use AMM in EDT

This section empirically discusses and demonstrates how to use AMM through experiments.

**Just adding AMM into the pre-trained model when inference** Table 8 shows when to use AMM in EDT. Model A is a pre-trained EDT-S* without AMM, getting an FID score of 50.8. Model B, which is added AMM from initialization and then trained, achieves an FID score of 52.1. Model C gains an FID score of 38.7, which is the model that added AMM to model A. The results show that using AMM during training will lead to poorer performance, and just adding AMM to the pre-trained model can greatly improve performance.

**Alternately inserting AMM into the decoder of EDT** We view the process of drawing pictures as a decoding process. So we think the decoder of EDT should be aligned with the act of humans drawing pictures. This inspires us where to place AMM in EDT. Namely, the alternation of attention range between global and local in drawing acts of humans should be imitated in the decoder of EDT. The AMM should be alternately or discontinuously inserted into the blocks of decoder.

To demonstrate the inspiration, in Figure 9, we try to compare Model A: pre-trained EDT-S* without AMM; Model B: alternately inserting AMM into encoder and decoder of pre-trained EDT-S*; Model C: consecutively inserting AMM into decoder of pre-trained EDT-S*; and Model D: alternately inserting AMM into the decoder of pre-trained EDT-S*. As shown in Table 9, Model D which is alternately inserted AMM into decoder of pre-trained EDT-S*, gets the lowest FID score.

When integrating AMM into a pre-trained model, the best arrangement of AMM in blocks varies across different models. Identifying the optimal placement and configuration of AMM requires testing and adjusting to realize its full potential.

### A.3.3   The determination of hyper-parameter

Table 10 shows the determination of hyper-parameter about the effective radius of local attention $R \in [0, d_{max}]$, where $d_{max} = \sqrt{2}(N - 1)$ is the farthest distance in a $N \times N$ token grid. In the table, when $R = \sqrt{(N-1)^2 + 4}$, EDT-S* achieves the lowest FID and highest Prec. So we set $R = \sqrt{(N-1)^2 + 4}$ in all sizes of model.

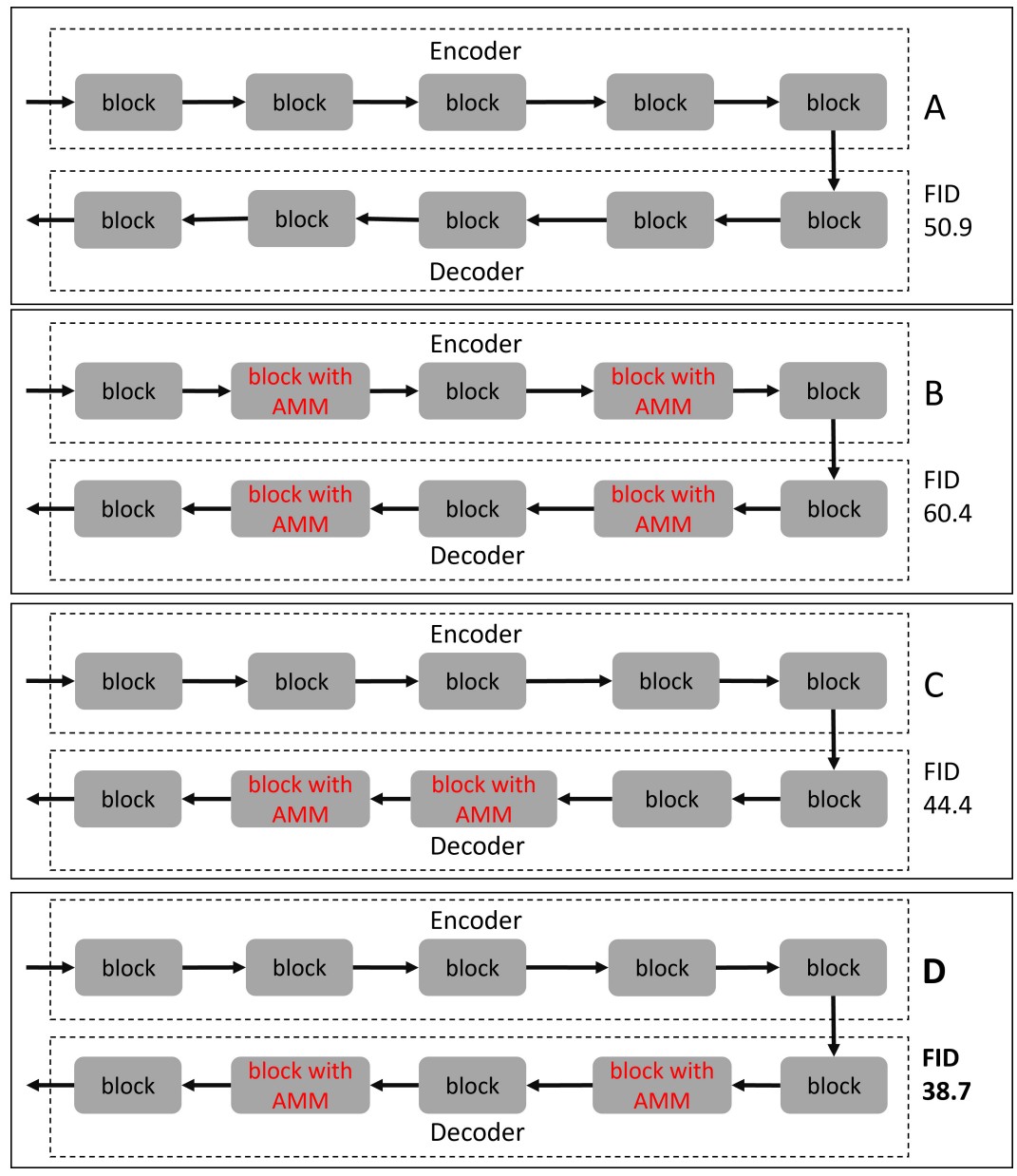

Figure 9: Different arrangement forms of AMM in EDT and their corresponding FID scores.

Table 10: Exploring the value of the effective radius of local attention in EDT-S* for $256 \times 256$ resolution.

| R | FID50K↓ | IS↑ | sFID↓ | Prec.↑ | Recall↑ |
|---|---|---|---|---|---|
| no AMM | 50.9 | 31.0 | 13.3 | 0.427 | 0.604 |
| $\sqrt{2}(N-1)$ | 38.9 | 36.6 | 9.3 | 0.472 | 0.626 |
| $\sqrt{(N-1)^2+4}$ | **38.7** | 36.4 | 9.3 | 0.473 | 0.623 |
| $\sqrt{(N-1)^2+1}$ | 39.0 | 36.5 | 9.3 | 0.474 | 0.621 |
| N-1 | 39.1 | 36.2 | 9.2 | 0.472 | 0.628 |
| $3N/4$ | 41.0 | 35.4 | 9.6 | 0.483 | 0.604 |
| $N/2$ | 47.4 | 32.1 | 11.7 | 0.495 | 0.583 |
| $N/4$ | 72.0 | 23.9 | 23.6 | 0.476 | 0.516 |

### A.3.4    The computational cost of AMM

The addition of AMM introduces minimal computational costs. Firstly, AMM can be incorporated into a pre-trained model without requiring additional fine-tuning, resulting in no additional training costs. Secondly, the increased computational cost of AMM during inference is negligible. For instance, in the last block of EDT-XL, the attention score matrix and the Attention Modulation Matrix are both 18×256×256 tensors. The computational cost of the Hadamard product between the attention score matrix and AMM is only 1.18M FLOPs for multiplication calculations, out of a total of 2819.3M FLOPs for the block. This amounts to merely 0.04% of the total FLOPs, making the computational cost of AMM negligible. In our experiments, we added 4 AMM modules to a pre-trained DiT-XL model, and the FID score decreased from 18 to 14.

### A.4    Discussion about token relation-enhanced masking training strategy

#### A.4.1    Analysis of the loss of EDT and MDT

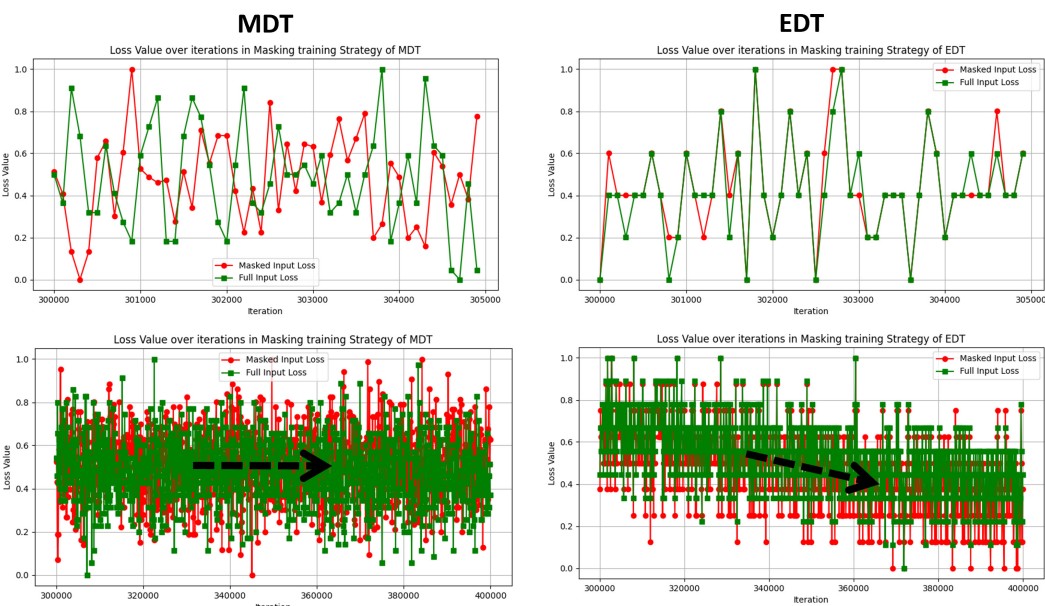

Figure 10: Comparing the loss changes of different masking training strategies.

In Figure 10, we separately applied the masking training strategies of MDT and EDT to train EDT-S and extracted $L_{masked}$ and $L_{full}$ values at the $300k \sim 305k$ and $300k \sim 400k$ training iterations. The left-top of the figure depicts the loss changes when using MDT's masking training strategy. As $L_{full}$ decreases, $L_{masked}$ increases, and vice versa, illustrating the conflict between these two losses. This conflict arises because $L_{masked}$ in MDT causes the model to focus on masked token reconstruction while ignoring diffusion training. As shown in the bottom-left of the figure, both the $L_{full}$ and $L_{masked}$ hardly decreased during the 300k to 400k training iterations. The right side of the figure shows the loss changes when using EDT's masking training strategy. The $L_{masked}$ and $L_{full}$ exhibit synchronized changes, and the loss values continuously decrease during the 300k to 400k training iterations.

#### A.4.2    The determination of the mask ratio

We explore the mask ratio of our masking training strategy. There are two down-sampling modules in EDTs. So we have two positions to implement token masking. As shown in Table 11, we determine the masking ratio in the first down-sampling module by training EDT-S. The best masking ratio in the first down-sampling module is $0.4 \sim 0.5$.

Table 11: Mask Ratio in the first down-sampling module.

| Mask Ratio | FID50K↓ | IS↑ | sFID↓ | Prec.↑ | Recall↑ |
|---|---|---|---|---|---|
| $0.1 \sim 0.2$ | 51.3 | 31.1 | 13.7 | 0.434 | 0.612 |
| $0.2 \sim 0.3$ | 48.6 | 33.5 | 13.2 | 0.441 | **0.631** |
| $0.3 \sim 0.4$ | 47.3 | **34.9** | 13.2 | 0.442 | 0.622 |
| $0.4 \sim 0.5$ | **46.4** | 34.1 | **11.7** | **0.449** | 0.621 |
| $0.5 \sim 0.6$ | 48.8 | 33.4 | 13.5 | 0.431 | 0.623 |

Table 12: Mask Ratio in the second down-sampling module. (Based on the $0.4 \sim 0.5$ mask ratio in the first down-sampling module)

| Mask Ratio | FID50K↓ | IS↑ | sFID↓ | Prec.↑ | Recall↑ |
|---|---|---|---|---|---|
| $0.1 \sim 0.2$ | **45.4** | **34.8** | 13.2 | 0.440 | **0.621** |
| $0.2 \sim 0.3$ | 46.2 | 34.6 | 13.5 | 0.441 | 0.619 |
| $0.3 \sim 0.4$ | 46.5 | 34.7 | 13.2 | 0.438 | 0.620 |
| $0.4 \sim 0.5$ | 47.2 | 34.4 | 13.4 | 0.432 | 0.620 |

Table 13: The comparison with existing SOTA methods on class-conditional image generation with classifier-free guidance on ImageNet 256×256 (CFG=2 in EDT; according to DiT and MDTv2, their optimal CFG settings are 1.5 and 3.8, respectively).

| Model | Cost↓ (Iter×BS) | GFLOPs↓ | FID↓ |
|---|---|---|---|
| DiT-S-G | 400K×256 | 6.06 | 21.03 |
| MDTv2-S-G | 400K×256 | 6.07 | 15.62 |
| **EDT-S-G(our)** | 400K×256 | **2.66** | **9.89** |
| ADM-G[4] | 1980k×256 | 1120.00 | 4.59 |
| LDM-4-G[11] | 178k×1200 | 104.00 | 3.60 |
| DiT-XL-G | 400K×256 | 118.64 | 5.50 |
| DiT-XL-G[21] | 7000K×256 | 118.64 | 2.27 |
| MDTv2-XL-G[33] | 4600K×256 | 118.69 | **1.58** |
| EDT-XL-G(our) | 400K×256 | **51.83** | 4.65 |
| EDT-XL-G(our) | 1000K×256 | **51.83** | 4.30 |
| EDT-XL-G(our) | 2000K×256 | **51.83** | 3.54 |

At the base of $0.4 \sim 0.5$ mask ratio in the first down-sampling module, we then determine the mask ratio in the second down-sampling module as shown in Table 12. According to the results, the best masking ratio in the second down-sampling module is $0.1 \sim 0.2$, at the base of $0.4 \sim 0.5$ mask ratio in the first down-sampling module.

## A.5 Additional Results

### A.5.1 Image generation with classifier-free guidance on ImageNet 256×256

The result of image generation with classifier-free guidance on ImageNet 256×256 is shown in Table 13. Under the same training cost, EDT-S-G achieves the lowest FID score compared to MDTv2-S-G (9.89 vs. 15.62). EDT-XL-G achieves a good balance in training cost, inference GFLOPs, and image generation performance.

### A.5.2 Comprehensive evaluation of AMM

**Using AMM under different iterations** We report the FID of EDT* with and without AMM under different iterations in Table 14. The results show that all EDTs* with AMM obtain lower FID.

Table 14: FID of EDTs* under different iterations on Imagenet $256 \times 256$.

| Iterations | EDT-S* no AMM | EDT-S* AMM | EDT-B* no AMM | EDT-B* AMM | EDT-XL* no AMM | EDT-XL* AMM |
|---|---|---|---|---|---|---|
| 50k | 96.4 | **80.8** | 79.5 | **64.8** | 50.1 | **45.8** |
| 100k | 64.5 | **58.5** | 46.0 | **40.4** | 23.7 | **21.3** |
| 150k | 58.2 | **51.0** | 39.7 | **32.6** | 17.9 | **15.3** |
| 200k | 55.8 | **47.0** | 37.3 | **29.0** | 15.6 | **12.6** |
| 250k | 53.3 | **44.0** | 35.5 | **26.9** | 14.6 | **11.4** |
| 300k | 52.4 | **42.0** | 34.2 | **25.1** | 14.1 | **10.7** |
| 350k | 51.2 | **40.1** | 33.5 | **24.2** | 14.5 | **10.6** |
| 400k | 50.9 | **38.7** | 33.2 | **23.2** | 14.9 | **10.5** |

Table 15: Results on various models across different sizes with (w) AMM and without (w/o) AMM on ImageNet.

| Model | Resolution | FID↓ | IS↑ | sFID↓ | Prec.↑ | Recall↑ |
|---|---|---|---|---|---|---|
| EDT-S*(w/o) | $256 \times 256$ | 50.9 | 31.0 | 13.3 | 0.427 | 0.604 |
| EDT-S* | $256 \times 256$ | **38.7** | **36.4** | **9.2** | **0.474** | **0.620** |
| EDT-S(w/o) | $256 \times 256$ | 46.9 | 35.4 | 13.5 | 0.442 | **0.624** |
| EDT-S | $256 \times 256$ | **34.3** | **42.6** | **13.1** | **0.501** | 0.612 |
| EDT-S(w/o) | $512 \times 512$ | 55.7 | 28.9 | 12.8 | 0.513 | **0.586** |
| EDT-S | $512 \times 512$ | **51.8** | **29.9** | **7.9** | **0.563** | 0.572 |
| EDT-B*(w/o) | $256 \times 256$ | 33.2 | 50.0 | 10.5 | 0.512 | **0.648** |
| EDT-B* | $256 \times 256$ | **23.2** | **62.3** | **8.9** | **0.573** | 0.624 |
| EDT-B(w/o) | $256 \times 256$ | 26.3 | 64.5 | 10.3 | 0.544 | **0.659** |
| EDT-B | $256 \times 256$ | **19.2** | **74.4** | **9.9** | **0.586** | 0.639 |
| EDT-XL*(w/o) | $256 \times 256$ | 14.9 | 96.5 | **8.0** | 0.617 | **0.667** |
| EDT-XL* | $256 \times 256$ | **10.5** | **117.8** | 9.9 | **0.663** | 0.637 |
| EDT-XL(w/o) | $256 \times 256$ | 12.8 | 111.7 | 8.2 | 0.627 | **0.685** |
| EDT-XL | $256 \times 256$ | **7.5** | **142.4** | **7.4** | **0.684** | 0.648 |
| DiT-XL/2(w/o) | $256 \times 256$ | 18.5 | 71.3 | **6.1** | 0.641 | **0.632** |
| DiT-XL/2 | $256 \times 256$ | **14.7** | **83.9** | 10.5 | **0.720** | 0.511 |

**More types of evaluation indicators on EDT with AMM** The Table 15 shows more types of evaluation indicators comparing EDTs without AMM and with AMM. EDTs* means the EDT models that don't use masking training strategy. In the table, when using AMM, the FID, IS and Precision are all improved. Furthermore, AMM is versatile and can be adapted to various diffusion transformers. For instance, the performance of DiT-XL is improved by AMM (18.5 FID vs. 14.7 FID).

Generated images using DiT-XL **without AMM**     Generated images using DiT-XL **with AMM**

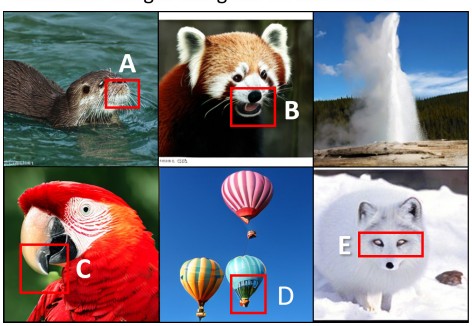 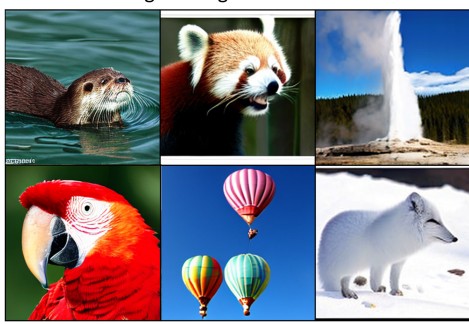

Figure 11: DiT-XL-400k with AMM achieves more realistic visual effects.

**Qualitative analysis on DiT-XL with AMM** We conduct qualitative analysis on DiT with and without AMM in Figure 11, which demonstrates AMM can also improve the local details of the synthesis images of DiT. The red boxes highlight the unrealistic areas in the images generated by DiT-XL without AMM. In the corresponding areas of the images generated by DiT-XL with AMM, the results appear more realistic. **Area A:** The otter lacks a mouth. **Area B:** The red panda's mouth is slightly distorted. **Area C:** The parrot's beak is not sharp enough. **Area D:** There are black stains on the right side of the hot air balloon. **Area E:** The fox's eyes are white. The steam image generated by DiT-XL without AMM is realistic. And the steam image generated by DiT-XL with AMM remains equally realistic. The addition of AMM does not negatively affect the original quality. The effectiveness of AMM on DiT-XL further demonstrates the universal applicability of AMM.

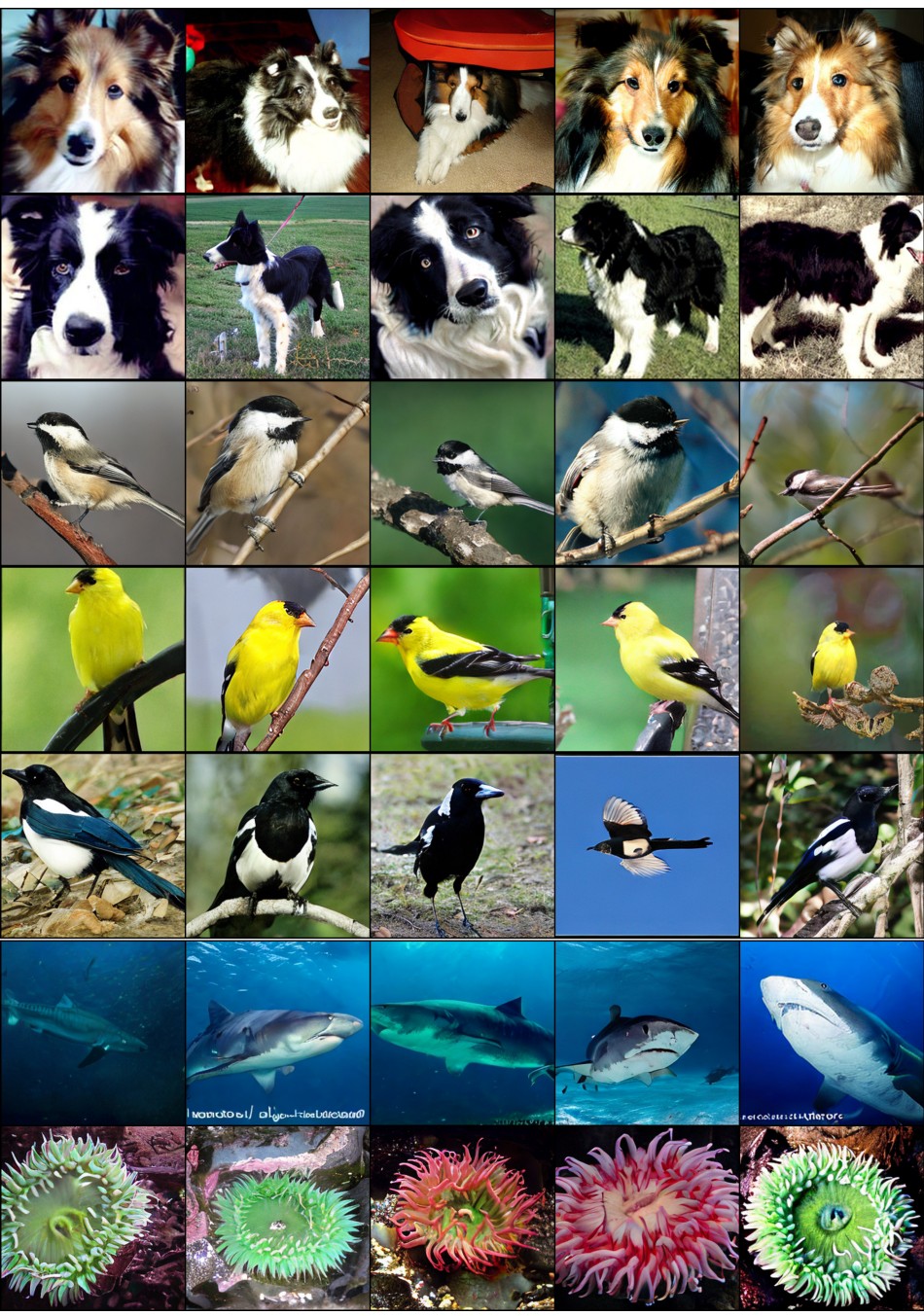

Figure 12: Visualization of images generated by the EDT-XL-2000K.

### A.5.3 Visualization

More visualized examples of EDT-XL-2000k generated images are shown in Figure 12. The sampling step is 250.

### A.6 Limitations

In this work, we propose the Efficient Diffusion Transformer (EDT) framework, which includes a lightweight-design of diffusion transformer, a training-free Attention Modulation Matrix (AMM), and its alternation arrangement in EDT inspired by human-like sketching and the token relation-enhanced masking training strategy. EDT effectively improves the training and inference speed of diffusion transformers. Notably, AMM is a new approach to improving diffusion models, with many aspects still worth exploring. When integrating AMM into a pre-trained model, the insertion and arrangement of AMM in blocks differ across various models. Thus, identifying the optimal placement and configuration of AMM requires testing to unlock its full potential. Moreover, the generation function of AMM still has room for improvement and deserves further exploration. For example, in addition to directly scaling attention scores by AMM, we can convert global attention into local attention during inference through methods like: (1) using attention window to exclude the interactions of tokens with far distance; (2) or scaling the attention based on token distance before the operation of softmax in attention modules. It is encouraged to further explore the idea of AMM, which incorporates local attention into transformers during inference, to improve other transformer-based models.

