# OpenReview forum: "EDT: An Efficient Diffusion Transformer Framework Inspired by Human-like Sketching"
_NeurIPS.cc/2024/Conference — NeurIPS 2024 poster_

### Official Review · Reviewer_MD5e · 2024-06-18

**Soundness:** 3
**Presentation:** 3
**Contribution:** 3
**Rating:** 6
**Confidence:** 1

**Summary:**

Transformer-based Diffusion Probabilistic Models (DPMs) have shown great potential in image generation tasks but are often hindered by extensive computational requirements. This paper introduces the Efficient Diffusion Transformer (EDT) framework to address these computational challenges. The EDT framework features a lightweight diffusion model architecture and incorporates a classifier-free Attention Modulation Matrix inspired by human-like sketching. Additionally, the authors propose a token relation-enhanced masking training strategy tailored explicitly for EDT to improve token relation learning. Extensive experiments demonstrate that the EDT framework significantly reduces both training and inference costs while surpassing the performance of existing transformer-based diffusion models in image synthesis. The paper highlights the effectiveness of the EDT framework in achieving a balance between computational efficiency and high performance in image generation tasks. The proposed methods and strategies demonstrate potential for broader applications and future research in the optimization of transformer-based models.

**Strengths:**

1. The paper introduces a novel way of approaching Transformer-based Diffusion Probabilistic Models, especially a classifier-free approach. The paper also attempts to understand the relationship between tokens, thereby making the pipeline efficient.
2. The pipeline is robust and lightweight. The paper introduces Attention Modulation matrix, that provides specific quantitative results for where to focus and how much to focus on an image. The authors have provided visual results on how this method has improved the quality of the images.

**Weaknesses:**

1. The pre-trained VAE might be trained on a classifier task, that might have some impact on categorization on certain classes.
2. The paper does not mention about the number of epochs or an estimate of time required by EDT to be able to produce this result.

**Questions:**

1. Is there any specific reason to use a VAE? There are other lightweight image feature extractor models available, that might make the pipeline even more efficient.
2. The experimental validation, while thorough, might benefit from a broader range of datasets to ensure the robustness and generalizability of the EDT framework across different image generation tasks.
3. Please add potential limitations of the work.

**Limitations:**

The paper is an extremely interesting read. The paper talks about the trade-off between computational resources and performance. However, there is no specific section dedicated, where the potential limitations have been discussed.

---

> ### Author Rebuttal · Authors · 2024-08-07
>
> Thanks for your positive feedback and insightful comments. We will answer your questions in the following:
>
> ---
>
> ***1. The usage of pre-trained VAE.***
> * **In the field of Latent Diffusion Models, pre-trained VAE is a commonly used model.** When training diffusion models or generating images, the pre-trained VAE remains frozen. The VAE encoder is used to encode images into latent representations and the VAE decoder is used to decode latent representations back into images.
>
> ---
>
> ***2. Training cost of EDT.***
>
> * We reported the training cost and speed **in Table 1 in the main paper**. For ImageNet 256×256, we estimated the training cost for EDT, MDTv2, and DiT on a 48GB-L40 GPU **in Rebuttal-Table 3 in the global rebuttal**, using a batch size of 256 and FP32 precision. **EDT achieves the best performance with a low training cost.** (GPU days refer to the number of days required for training on a single L40 GPU.)
>
> | Model|Epochs|Training images|GPU days| FID|
> |---|---|---|---|---|
> |**EDT-S** |80|102M|**2.75**|**38.73**|
> |DiT-S|80|102M|2.96|68.40|
> |MDTv2-S|80|102M|16.47|39.50|
> |**EDT-B**|80|102M|9.19|**19.18**|
> |DiT-B|80|102M|**8.62**|43.47|
> |MDTv2-B|80|102M|26.17|19.55|
> |**EDT-XL**|80|102M|**37.79**|**7.52**|
> |DiT-XL|80|102M|39.82|19.47|
> |MDTv2-XL|80|102M|72.62|7.70|
>
> **Rebuttal-Table 3**: The training cost and FID of EDT, DiT, and MDTv2 on ImageNet 256×256 with batch size of 256 and FP32 precision.
>
> ---
>
> ***3. Why use VAE.***
> * VAE is used to reduce the computational cost of diffusion models. The diffusion model generates images through multiple iterations, whose input in each iteration is its output in the previous iteration. To reduce the computational cost of each iteration, we make the diffusion model train and predict in latent space, since the feature size in latent space is smaller than that in the pixel space of images. During inference, the diffusion model is used to generate the latent representations of images, and the VAE decoder is used to decode latent representations back into images.
>
> ---
>
> ***4. Experiments on extra dataset celebA-HQ.***
> * We conducted a new experiment on CelebA-HQ 256×256 for the unconditional image synthesis task. We train EDT-S, DiT-S, and MDTv2-S on CelebA-HQ 256×256 under the same training epochs with the default settings in their papers, respectively. **As shown in Rebuttal Table 2, EDT-S achieved the lowest FID, demonstrating that EDT is also effective in unconditional image synthesis tasks.**
>
> |Model|Training images|FID|
> |---|---|---|
> |EDT-S|100k * 240|**16.60**|
> |DiT-S|100k * 240|19.12|
> |MDTv2-S|100k * 240|17.34|
>
> **Rebuttal-Table 2**: Evaluation of unconditional image synthesis on CelebA-HQ.
>
> ---
>
> ***5. Potential limitations.***
> * The experiments were conducted on class-conditional image synthesis tasks and unconditional image synthesis. Future work will explore other image synthesis tasks such as text-to-image tasks.
> * When integrating AMM into a pre-trained model, the insert and arrangement of AMM blocks varies across different models. Identifying the optimal placement and configuration of AMM requires testing to fully realize its potential.
> * Although significant progress has been made with AMM, the generation function of AMM still has room for improvement and warrants further exploration.

---

> > ### Comment · Reviewer_MD5e · 2024-08-08
> >
> > Thank you for the insightful discussion!

---

> > > ### Author Response · Authors · 2024-08-10
> > >
> > > Thank you for your review! Please let me know if you have any questions or concerns.

---

### Official Review · Reviewer_8pvq · 2024-06-29

**Soundness:** 3
**Presentation:** 2
**Contribution:** 2
**Rating:** 5
**Confidence:** 5

**Summary:**

This paper introduces the Efficient Diffusion Transformer (EDT) framework to address the high computational requirements of transformer-based Diffusion Probabilistic Models (DPMs). The EDT framework features a lightweight diffusion model architecture, a training-free Attention Modulation Matrix inspired by human sketching, and a token relation-enhanced masking training strategy. Extensive experiments show that EDT reduces training and inference costs while surpassing existing transformer-based diffusion models in image synthesis performance. For example, EDT-S achieves a lower FID score of 34.27 and offers significantly faster training and inference speeds compared to MDTv2-S.

**Strengths:**

+ The proposed network can reduce the computation of diffusion is a plus point although it requires more parameters compared to existing methods.

+ Some experimental results on ImageNEt 256x256 show promising looks.

**Weaknesses:**

**[Update]: after rebuttal, I raised score from 4 to 5**

The proposed method shows some good merits in the architectural design, however, it remains several major concerns:

+ Most results are reported with not very optimal settings. Specifically, classifier-free guidance is the standard practice, but it is omitted, and cannot show the advantage of the proposed method. This results in the not-very convincing evaluation that the proposed method is superior to the existing approaches with SOTAs (DiT, MDT, RDM [1], etc).

+ More datasets (or resolutions, such as 512) need to be conducted to have a more conclusive finding for the effectiveness of the proposed method as the proposed method is lightweight, it should not be a big deal with a higher resolution training.

[1] Relay Diffusion: Unifying diffusion process across resolutions for image synthesis, ICLR 2024

**Questions:**

My main concerns are listed in the weaknesses of its evaluation of the CFG setting and other datasets.

**Limitations:**

Yes, they discussed it.

---

> ### Author Rebuttal · Authors · 2024-08-07
>
> We thank the reviewer for the valuable feedback and suggestions. We will answer your questions in the following:
>
> ---
>
> ***1. The experiment under optimal CFG settings.***
>
> * According to DiT and MDTv2, their optimal CFG settings are 1.5 and 3.8, respectively. Based on our experimental exploration, the optimal Cfg for EDT is 2. **As shown in Rebuttal-Table 4**, when compared under their optimal CFG settings with EDT and at the same training cost, **EDT-S achieves the best performance with the lowest FID**.
>
> | Models|Training images|FID|
> |---|---|---|
> |EDT-S CFG=2|400k*256|**9.89**|
> |MDTv2-S CFG=3.8|400k*256|15.62|
> |DiT-S CFG=1.5|400k*256|21.03|
> |DiT-XL CFG=1.5|400k*256|5.50|
> |EDT-XL CFG=2|400k*256|**4.65**|
>
> **Rebuttal-Table 4**: Comparison and evaluation under optimal Cfg on ImageNet 256×256.
>
> ---
>
> ***2. Experiments on extra dataset CelebA-HQ.***
>
> * We conducted a new experiment on CelebA-HQ 256×256 for the unconditional image synthesis task. We train EDT-S, DiT-S, and MDTv2-S on CelebA-HQ 256×256 under the same training epochs with the default settings in their papers, respectively. **As shown in Rebuttal Table 2, EDT-S achieved the lowest FID, demonstrating that EDT is also effective in unconditional image synthesis tasks.**
> Due to the limited time, we did not finish the experiments with the 512x512 resolution training. We will include it in the revision.
>
> |Model|Training images|FID|
> |---|---|---|
> |EDT-S|100k * 240|**16.60**|
> |DiT-S|100k * 240|19.12|
> |MDTv2-S|100k * 240|17.34|
>
> **Rebuttal-Table 2**: Evaluation of unconditional image synthesis on CelebA-HQ.

---

> > ### Comment · Reviewer_8pvq · 2024-08-12
> >
> > Thanks for the rebuttal. It partly resolved my concerns, however, still remained. First, putting some numeric numbers is not very convincing. In the submission (including the appendix), the qualitative results (generated samples) are absent and it is very important to see the generation tasks to support their quantitative numbers.
> >
> > In addition to the missing comparison with other methods on 512 resolution, the author heavily referred to and compared with MDTv2 but they only focused on the earlier convergence of training steps. In MDTv2 they compared with baseline DiT on both earlier (e.g. 400k steps) and full convergence (4600k steps) and MDTv2 reached their best FID 1.58 on ImageNet 256x256.
> > The authors claim the efficiency of the method with much lower computation cost, which raises the question of why they ignored their optimal and final results as their models are trained until fully converged. Can it beat the best FID of MDTv2?
> >
> > Qualitative results comparison to support their numerical number should also be carefully prepared for both their optimal generated images and the comparison of reconstructed images using the masking strategy of MDT and their masking strategy (see MDTv1 ICCV version for the referred image reconstruction with masking). I expect that the reconstructed output of the proposed method with their masking is better than MDT.

---

> > > ### Author Response · Authors · 2024-08-13
> > >
> > > We sincerely thank you for your valuable suggestions regarding the comparative experiments and qualitative analysis of the masking training strategy of EDT. These additions make the results more understandable and convincing than relying solely on numerical data.
> > >
> > > ---
> > >
> > > ***1. Qualitative results comparison of inpainting images using models trained by the masking training strategy of MDT and EDT.***
> > >
> > > After following your advice regarding the qualitative results of masking training and referring to MDTv1, we conducted inpainting experiments using EDT-S under different masked ratios. We utilized two versions of the pre-trained EDT-S models, each employing the masking training strategy of MDT and EDT, respectively. We applied various mask ratios to the images and used these two models to inpaint the masked areas. **When the mask ratio reached 50%, the EDT-S model using MDT's masking strategy struggled to reconstruct the images, while the EDT-S model using EDT's masking strategy was still able to do so.** We plan to include this visual comparison in the appendix.
> > >
> > > ---
> > >
> > > ***2. The experiment of EDT-XL for full convergence.***
> > >
> > > Our primary focus has been on the efficiency of diffusion transformers. Our proposed EDT offers the community a more efficient diffusion model architecture, which achieves speed-ups of 2.29x, 2.29x, and 2.22x in inference speed on small, base, and xlarge sizes respectively. Unfortunately, training an EDT-XL for 4600k iterations would require approximately two months for one round of training, and we could not complete this due to resource limitations. Nonetheless, we have validated that EDT maintains acceptable accuracy while significantly improving speed within our capacity. **Training diffusion model is exceptionally costly, and we are currently training an EDT-XL for 2000k iterations, with plans to include the relevant results in the updated version of the paper.**
> > >
> > > ---
> > >
> > > ***3. Experiment on different datasets to evaluate the effectiveness of the EDT.***
> > >
> > > Unfortunately, because of time and resource limitations, we were unable to conduct experiments at 512 resolution. Instead, we performed experiments on a different dataset (CelebA-HQ), where EDT-S showed competitive results.
> > >
> > > ---
> > >
> > > In addition, please note that the masking training strategy in our work is just one part of what we have contributed. It aims to mitigate the performance loss of the efficient diffusion transformer architecture. **Our contributions also encompass the model architecture design and the AMM plugin. These modules are not proposed by the current DiT works, and we believe they can offer new insights for developing this field.**

---

> > > > ### Comment · Reviewer_8pvq · 2024-08-13
> > > >
> > > > Thank you for the response; it seems to address my concerns. However, I'm curious why the comparison images weren't provided now but waited to be put into the Appendix. Could the authors upload the qualitative images to a free platform like https://postimg.cc/ and share the link here, ensuring anonymity? I've seen this link used in conferences like ICLR and ICML for reviews and rebuttals. From what I've quickly read, NeurIPS policy seems to allow external links. See [1] for reference to sharing images.
> > > >
> > > > [1] Relay Diffusion: Unifying diffusion process across resolutions for image synthesis, ICLR 2024.

---

> ### Author Response · Authors · 2024-08-14
>
> Thank you for your helpful suggestions and for providing a method for submitting qualitative images. We have submitted two sets of qualitative images at [https://postimg.cc/gallery/ZS1FXcb](https://postimg.cc/gallery/ZS1FXcb).
>
> ---
>
> ***Figure 1 is a qualitative analysis of the AMM plugin.***
>
> AMM is a train-free plugin that can be inserted into pre-trained models to improve image generation performance. We compared images generated by EDT-XL with and without AMM to qualitatively analyze the effect of AMM on image generation. In Figure 1, the red boxes highlight the unrealistic areas in the images generated by EDT-XL without AMM. **In the corresponding areas of the images generated by EDT-XL with AMM, the results appear more realistic.** Moreover, the parrot image generated by EDT-XL without AMM is realistic and the parrot image generated by EDT-XL with AMM still remains equally realistic. Therefore, **adding AMM does not negatively affect the original quality.** This visual analysis demonstrates the effectiveness of the AMM plugin.
>
> ---
>
> ***Figure 2 is a qualitative results comparison of reconstructing masked images using models trained by the masking training strategy of MDT and EDT.***
>
> We conducted inpainting experiments using EDT-S under different masked ratios. We utilized two versions of the pre-trained EDT-S models, each employing the masking training strategy of MDT and EDT, respectively. In Figure 2, We applied various mask ratios to the images and used these two models to reconstruct the masked areas. **When the mask ratio reached 50%, the EDT-S model using MDT's masking strategy struggled to reconstruct the images, while the EDT-S model using EDT's masking strategy was still able to do so. EDT trained by EDT's masking training strategy demonstrates better image reconstruction capabilities.** This visual analysis demonstrates the effectiveness of the EDT's masking training strategy.

---

> > ### Author Response · Authors · 2024-08-14
> >
> > Please note that we have highlighted the difference between the samples, as suggested by reviewer FhNv, and edited the comment above.

---

### Official Review · Reviewer_QU5R · 2024-07-10

**Soundness:** 3
**Presentation:** 3
**Contribution:** 3
**Rating:** 5
**Confidence:** 4

**Summary:**

The Efficient Diffusion Transformer (EDT) framework is developed, featuring a lightweight architecture designed based on thorough computational analysis. Inspired by human sketching, EDT alternates between global attention and location attention. Additionally, the Attention Modulation Matrix enhances the detail of generated images in pre-trained diffusion transformers without requiring extra training.
 A novel token masking training strategy is proposed to improve the token relation learning capability of EDT. EDT achieves a new state-of-the-art performance and faster training and inference speeds compared to existing models like DiT and MDTv2. A series of exploratory experiments and ablation studies were conducted to analyze and identify the key factors affecting EDT's performance.

**Strengths:**

Novelty: The three innovative techniques introduced for designing efficient diffusion transformers represent a significant advancement in the field. These techniques, which include the development of a lightweight architecture, the introduction of the Attention Modulation Matrix, and a novel token masking training strategy, are unique contributions that set this work apart from existing approaches.

Significance: The challenge of designing efficient diffusion transformers for training and inference is a critical issue in the development of diffusion models. The work’s focus on enhancing efficiency without compromising performance is particularly relevant in the context of growing model sizes and the demand for faster computational methods.

Methodology: The proposed algorithm is well formulated and clearly explained. The approach includes a comprehensive computational analysis to design a lightweight diffusion transformer architecture, inspired by human sketching with an alternation process between global attention and location attention. The introduction of the Attention Modulation Matrix is particularly noteworthy, as it improves image detail in pre-trained diffusion transformers without additional training costs. The novel token masking training strategy enhances the learning ability of the model, demonstrating a sophisticated understanding of token relationships.

Results: The experimental results show improvements over existing methods such as DiT and SD-DiT.  A series of exploratory experiments and ablation studies further validate the robustness and effectiveness of the proposed techniques, providing a detailed analysis of the key factors influencing EDT's performance.

**Weaknesses:**

1. The training dataset is limited to ImageNet, which may not fully represent the broader applicability of the method.
2. The Attention Modulation Matrix (AMM) could be tested on text-to-image models, such as the Pixart Series, to demonstrate the wider applicability of the proposed method.
3. The motivation behind the lightweight design is not entirely clear. It seems to balance between UNet and transformer architectures but appears more similar to the UNet.
4. The computational process of the AMM would benefit from additional figures or diagrams to enhance understanding.

**Questions:**

1. Why do long skip connection modules inevitably lead to a loss of token information? From my perspective, long skip connections represent a fundamental difference between UNet and transformer architectures, potentially enhancing the training process by facilitating information flow across layers. If these connections are perceived as a drawback, wouldn't it be more effective to adopt the vanilla transformer architecture, which typically avoids such connectivity patterns?

2. What is the additional computational cost incurred by integrating the Attention Modulation Matrix into pre-trained DiT models? Understanding the impact on computational resources is crucial for assessing the feasibility and scalability of implementing this enhancement across different model configurations and training scenarios.

3. Can you provide a detailed comparison of the training costs associated with your proposed method versus other relevant approaches? Analyzing the computational overhead, training time, and resource requirements relative to existing methods will provide valuable insights into the practical advantages and trade-offs of adopting your approach for training and deploying diffusion transformers.

**Limitations:**

The evaluation and application of the proposed method is limited.

---

> ### Author Rebuttal · Authors · 2024-08-07
>
> Thanks for your insightful comments and suggestions. We will address your concerns in the following answers:
>
> ---
>
> ***1. Experiments on extra dataset CelebA-HQ.***
>
> * We conducted a new experiment on CelebA-HQ 256×256 for the unconditional image synthesis task. We train EDT-S, DiT-S, and MDTv2-S on CelebA-HQ 256×256 under the same training epochs with the default settings in their papers, respectively. **As shown in Rebuttal Table 2, EDT-S achieved the lowest FID, demonstrating that EDT is also effective in unconditional image synthesis tasks.**
>
> |Model|Training images|FID|
> |---|---|---|
> |EDT-S|100k * 240|**16.60**|
> |DiT-S|100k * 240|19.12|
> |MDTv2-S|100k * 240|17.34|
>
> **Rebuttal-Table 2**: Evaluation of unconditional image synthesis on CelebA-HQ.
>
> ---
>
> ***2.Motivation for lightweight design.***
>
> * **Our motivation for a lightweight design is to decrease the computational overhead by reducing the number of tokens in the intermediate blocks.** We achieve this by using down-sampling modules to compress the tokens. This idea is similar to U-Net, but we implemented it to the transformer-based diffusion models. **However, while compressing tokens reduces the computational overhead of EDT, it also compromises token information. To address this issue, we have enhanced the down-sampling modules and long skip connection modules,** incorporating techniques such as ''token information enhancement'' and ''positional encoding supplement''. **As shown in Table 4 of the Appendix,** the effectiveness of these improvements is demonstrated through ablation experiments.
>
> ---
>
> ***3. The computational process of the AMM.***
>
> * **In Rebuttal-Figure 2 in the PDF of global rebuttal**, we illustrate the process of attention modulation. Image can be split into N patches and each token is the feature of a patch.  Each token (patch) corresponds to a rectangular area of the image and has a corresponding 2-D coordinate (x, y) in the image. We calculate an Euclidean distance value $d$ for each pair of tokens, resulting in a distance matrix, which is an N×N tensor. Based on the distance matrix, we generate modulation values $m$ via the modulation matrix generation function $F(d)$, which assigns lower modulation values to tokens that are farther apart. These modulation values form an Attention Modulation Matrix (AMM), another N×N tensor. **Importantly, we integrate the AMM into the pre-trained EDT without any additional training.** The attention modulation matrix is calculated when the model is instantiated. During inference, the modulated attention score matrix is obtained by performing a Hadamard product between the attention modulation matrix and the attention score matrix.
>
> ---
>
> ***4. The computational cost of AMM.***
>
> * **The addition of AMM introduces minimal computational costs.** Firstly, AMM can be incorporated into a pre-trained model without requiring additional fine-tuning, resulting in no additional training costs. Secondly, the increased computational cost of AMM during inference is negligible. For instance, in the last block of EDT-XL, the attention score matrix and the Attention Modulation Matrix are both 18×256×256 tensors. The computational cost of the Hadamard product between the attention score matrix and AMM is only 1.18M FLOPs for multiplication calculations, out of a total of 2819.3M FLOPs for the block. **This amounts to merely 0.04% of the total FLOPs, making the computational cost of AMM negligible.** In our experiments, we added 5 AMM modules to a pre-trained DiT-XL model, and the FID score decreased from 18 to 14.
>
> ---
>
> ***5. The loss of token information in long skip connection.***
>
> * Long skip connection modules are crucial for the performance of diffusion models, but the token merging operations within these modules can result in the potential loss of token information. Before the long skip connection, we have two sets of tokens, each represented as an N × D tensor. During the long skip connection, these two sets of tokens are concatenated into an N × 2D tensor and then merged into an N × D tensor through a linear layer. **This dimensionality reduction from 2D to D leads to a loss of token information and disrupts positional information.** To address this issue, we introduced Token Information Enhancement and Positional Encoding Supplement within the long skip connections, as demonstrated **in Table 4 of the Appendix**.
>
> ---
>
> ***6. Training cost of EDT.***
>
> * We reported the training cost and speed **in Table 1 in the main paper**. For ImageNet 256×256, we estimated the training cost for EDT, MDTv2, and DiT on a 48GB-L40 GPU **in Rebuttal-Table 3 in the global rebuttal**, using a batch size of 256 and FP32 precision. **EDT achieves the best performance with a low training cost.** (GPU days refer to the number of days required for training on a single L40 GPU.)
>
> | Model|Epochs|Training images|GPU days| FID|
> |---|---|---|---|---|
> |**EDT-S** |80|102M|**2.75**|**38.73**|
> |DiT-S|80|102M|2.96|68.40|
> |MDTv2-S|80|102M|16.47|39.50|
> |**EDT-B**|80|102M|9.19|**19.18**|
> |DiT-B|80|102M|**8.62**|43.47|
> |MDTv2-B|80|102M|26.17|19.55|
> |**EDT-XL**|80|102M|**37.79**|**7.52**|
> |DiT-XL|80|102M|39.82|19.47|
> |MDTv2-XL|80|102M|72.62|7.70|
>
> **Rebuttal-Table 3**: The training cost and FID of EDT, DiT, and MDTv2 on ImageNet 256×256 with batch size of 256 and FP32 precision.
>
> ---
>
> ***7. Adding AMM to Text-to-Image Models.***
> * Due to time constraints, incorporating AMM into text-to-image models will be addressed in our future work.

---

> ### Author Response · Authors · 2024-08-14
> **Additional qualitative results for both AMM and masking training strategy**
>
> As reviewers FhNv and 8pvq suggested, we added qualitative results to visualize our results.
>
> ---
>
> We highlight the different areas by the red box on the samples. Please refer to the new Link at [https://postimg.cc/gallery/ZS1FXcb](https://postimg.cc/gallery/ZS1FXcb).
>
> ---
>
> ***1.The difference between the provided samples generated by the models with/without AMM.***
>
> In Figure 1, the red boxes highlight the unrealistic areas in the images generated by EDT-XL without AMM. **In the corresponding areas of the images generated by EDT-XL with AMM, the results appear more realistic.** Moreover, the parrot image generated by EDT-XL without AMM is realistic and the parrot image generated by EDT-XL with AMM still remains equally realistic. Therefore, **adding AMM does not negatively affect the original quality.** This visual analysis demonstrates the effectiveness of the AMM plugin.
>
> ---
>
> ***2.The difference between the provided samples generated by the models with the proposed masking training strategy and MDT's masking strategy.***
>
> In Figure 2, We applied various mask ratios to the images and used these two models to reconstruct the masked areas during inference. **When the mask ratio reached 50%, the EDT-S model using MDT's masking strategy struggled to reconstruct the images, while the EDT-S model using EDT's masking strategy was still able to do so.** EDT trained by EDT's masking training strategy demonstrated better image reconstruction capabilities. This visual analysis demonstrates the effectiveness of the EDT's masking training strategy.
>
> ---
>
> Thank you for your valuable and insightful suggestions. We are looking forward to hearing your feedback on EDT. If you have any questions, we are more than willing to provide further clarification and address any issues promptly.

---

### Official Review · Reviewer_FhNv · 2024-07-14

**Soundness:** 3
**Presentation:** 2
**Contribution:** 3
**Rating:** 5
**Confidence:** 5

**Summary:**

The paper introduces a new efficient diffusion-based model, namely EDT. First, they revisit the masking strategy proposed in MDT and provide some insights regarding the discrepancy in the training objective. To this end, EDT uses a more efficient masking mechanism that focuses on the main generation task instead of paying more attention to recovering the masked regions.
Secondly, the main design is inspired by the human brain, which is nice, where they implement an alternating mechanism to alternate between local and global attention.

**Strengths:**

* Alternating between the local and global details inspired by human brains is novel.
* Tackling an important application.
* The paper's writing is good, which makes the paper easy to follow in general.

**Weaknesses:**

* [Methodolgy] The papers tackle an important application. However, I am concerned about the execution way. Designing architecture inspired by human brains is very important; however, the gain seems limited.
* [Expirements] It is mentioned in lines 43, 62, and 193 that the AMM could be easily integrated into any existing method. However, there are no experiments to support this claim. This is crucial to show the effectiveness of the proposed module.
* [Expirements] Conducting the experiments on only one dataset is insufficient. It is recommended to show the gain of the proposed method on Celeb-HQ and Lsun-Churches datasets. I suggested these two as their scale is considerably small thus conducting these experiments should not be challenging.
* [Results] The claims in the abstract related to the efficiency by saying your method achieves 4 x speed up in training is misleading. Despite your clarification, this is only valid for the small variant, still I was expecting significant improvements on the other variants as well, which is not the case.
* [Analysis] The analysis regarding the MVD-v2 masking mechanism is shallow. In addition, presenting the loss trends using screenshots of the terminal is shocking.
* [Ablations] Ablation studies are very important so differing them to the appendix could be problematic. I recommend saving some space and including the ablations in the main paper.
* [Visualization] All the Figures must be at the top of the pages to enhance readability.
* [Experiments] The related work is too short and not enough. At least, I was expecting a longer version in the appendix. In addition, some important work related to this is missing. I would suggest the author include Toddler [1] as its motivation, designing an efficient diffusion-based generative model inspired by human brains, which is very similar.

[1] Bakr, Eslam Mohamed, et al. "ToddlerDiffusion: Flash Interpretable Controllable Diffusion Model." arXiv preprint arXiv:2311.14542 (2023).

**Questions:**

* Please show the effectiveness of your method on more datasets, as suggested earlier.
* Integrate the AMM module into different methods to show its effectiveness as claimed.

**Limitations:**

* The limitations are not discussed.

---

> ### Author Rebuttal · Authors · 2024-08-07
>
> We appreciate the professional and insightful comments. We address each comment as follows:
>
> ---
>
> ***1. Performance improvement by AMM.***
>
>
> - We demonstrate that AMM is both effective and efficient through several experiments detailed **in Tables 9, 10, and 11 in the Appendix**. Tables 9 and 10 compare model performance with and without AMM across different sizes of EDTs. **Models incorporating AMM achieve lower FID scores and higher IS scores**. For example, Table 9 shows that EDT-S with AMM achieves a lower FID score of 34.2 compared to 46.9 for EDT-S without AMM. **Notably, AMM is integrated into pre-trained EDT-S without additional training**. Table 11 illustrates AMM's effectiveness across different iterations of EDT. Furthermore, AMM and its arrangement in blocks are designed by mimicking the brain's logical process—alternating between global and local attention as observed in human sketching. We will include a detailed ablation study of AMM in the main paper.
>
> ---
>
> ***2. More models incorporated with AMM.***
> - We integrated AMM into pre-trained EDT, DiT, and MDT, which are transformer-based models. We conducted experiments on ImageNet and Celeba-HQ **in Rebuttal-Table 1**. **The models with AMM obtain lower FID compared to the models without AMM**, which further demonstrates the effectiveness of AMM in different models and datasets.
>
> |**On ImageNet 256×256 (class-conditional image synthesis)**||||
> |---|---|---|---|
> |Model|Training images|W/o AMM|W AMM|
> |EDT-S|400k * 256|42.60|**34.20**|
> |DiT-S|400k * 256|67.16|**63.11**|
> |MDTv2-S|400k * 256|39.02|**31.89**|
> |EDT-XL|400k * 256|12.80|**7.52**|
> |DiT-XL|400k * 256|18.48|**14.73**|
> |DiT-XL|7000k * 256|9.62|**3.75**|
> |**On CelebA-HQ 256 × 256 (unconditional image synthesis)**||||
> |Model|Training images|W/o AMM |W AMM|
> |EDT-S|100k * 240|17.01|**16.60**|
> |DiT-S|100k * 240|19.12|**18.41**|
> |MDTv2-S|100k * 240|17.34|**17.11**|
>
> **Rebuttal-Table 1**: Evaluation of the performance of pre-trained EDT, DiT, and MDT without and with AMM on ImageNet and CelebA-HQ.
>
> ---
>
> ***3. Experiments on extra dataset CelebA-HQ.***
> - We conducted a new experiment on CelebA-HQ 256×256 for the unconditional image synthesis task. We train EDT-S, DiT-S, and MDTv2-S on Celeb-HQ 256×256 under the same training epochs with the default settings in their papers, respectively. **As shown in Rebuttal Table 2, EDT-S achieved the lowest FID, demonstrating that EDT is also effective in unconditional image synthesis tasks.**
>
> |Model|Training images|FID|
> |---|---|---|
> |EDT-S|100k * 240|**16.60**|
> |DiT-S|100k * 240|19.12|
> |MDTv2-S|100k * 240|17.34|
>
> **Rebuttal-Table 2**: Evaluation of unconditional image synthesis on CelebA-HQ.
>
> ---
>
> ***4. Clarification of the speed-ups in training.***
> - As shown **in Table 1 of the main paper**, EDT-S, EDT-B, and EDT-XL attain speed-ups of 3.93x, 2.84x, and 1.92x respectively in the training phase, and 2.29x, 2.29x, and 2.22x respectively in inference, when compared to the corresponding sizes of MDTV2. We will provide a more explicit description regarding the speed-up details.
>
> ---
>
> ***5. Analysis of masking training mechanism.***
> - By observing the loss changes **in Rebuttal-Figure 1 in the PDF of the global rebuttal**, we identified a conflict between $L_{masked}$ (loss when the input consists of the remaining tokens after masking) and $L_{full}$ (loss when the input consists of the full token input) in MDTv2. Although MDTv2 also observed similar issues, they did not solve them effectively.
> In MDTv2, it was found that using only $L_{masked}$ caused the model to overly focus on reconstructing the masked tokens, thereby neglecting diffusion training. To address this, both the full token input and the masked token input were fed to the diffusion model, resulting in the inclusion of both $L_{masked}$ and $L_{full}$ in MDTv2's loss function.
> However, our work discovered a conflict between $L_{masked}$ and $L_{full}$. We separately applied the masking training strategies of MDTv2 and EDT to train diffusion models and extracted $L_{masked}$ and $L_{full}$ values at the 300k~305k training iterations. **As shown in Rebuttal-Figure 1 in the PDF of the global rebuttal**, we visualized the changes of $L_{masked}$ and $L_{full}$. **The left-top of Rebuttal-Figure 1** depicts the loss changes when using MDTv2's masking training strategy. As $L_{full}$ decreases, $L_{masked}$ increases, and vice versa, illustrating the conflict between these two losses. This conflict arises because $L_{masked}$ causes the model to focus on masked token reconstruction while ignoring diffusion training. **As shown in the bottom-left of Rebuttal-Figure 1, both the $L_{full}$ and $L_{masked}$ hardly converged during the 300k to 400k training iterations.**
> In our work, to resolve this conflict, we performed masking in the intermediate layer instead of before input. This method forces the model learning to establish relations between tokens before they are masked. **The right side of Rebuttal-Figure 1** shows the loss changes when using EDT's masking training strategy. With EDT's strategy, $L_{masked}$ and $L_{full}$ exhibit synchronized changes, and **the loss values continuously decrease during the 300k to 400k training iterations.** Due to the page limit, the aforementioned discussion will be included in the supplementary material for the camera-ready version.
>
> ---
>
> ***6. Ablations.***
> - We will rearrange the work and place the ablations table in the main paper.
>
> ---
>
> ***7. Visualization.***
> - We will make all figures be placed at the top of the pages to enhance readability.
>
> ---
>
> ***8. Related work***
> - We will improve our related work and provide more comprehensive writing in the appendix, and we will discuss important work such as Toddler in the related work.

---

> > ### Comment · Reviewer_FhNv · 2024-08-14
> > **Thanks to Authors**
> >
> > I appreciate the author's rebuttal and efforts.
> > After carefully reading the other reviewers' rebuttals and comments, I am more inclined to raise my score to 5 instead of 4.
> >
> > As I have some concerns, as follows:
> > - The qualitative results in response to the reviewer "8pvq" do not show the superiority of the proposed methods. I see no difference between the provided samples.
> > - The quantitative results show some gains but are still not so convincing. For instance, the reported FID on CelebHQ is too high. The diffusion-based models, even the small ones, can easily get around 7 FID on CelebHQ.
> > - I also agree with other reviewers about the importance of showing the effectiveness of the proposed method on higher resolution, e.g., 512. However, I also understand the hardware challenges; thus, I will not put any weight on this.

---

> ### Author Response · Authors · 2024-08-14
>
> Thank you for your professional and insightful comments. These comments enhance the quality of our work. We appreciate the recognition of our efforts to improve the score. Below are our responses to the follow-up questions.
>
> ---
>
> We highlight the different areas by the red box on the samples. Please refer to the new Link at [https://postimg.cc/gallery/ZS1FXcb](https://postimg.cc/gallery/ZS1FXcb).
>
> ---
>
> ***1.The difference between the provided samples generated by the models with/without AMM.***
>
> In Figure 1, the red boxes highlight the unrealistic areas in the images generated by EDT-XL without AMM. **In the corresponding areas of the images generated by EDT-XL with AMM, the results appear more realistic.** Moreover, the parrot image generated by EDT-XL without AMM is realistic and the parrot image generated by EDT-XL with AMM still remains equally realistic. Therefore, **adding AMM does not negatively affect the original quality.** This visual analysis demonstrates the effectiveness of the AMM plugin.
>
> ---
>
> ***2.The difference between the provided samples generated by the models with the proposed masking training strategy and MDT's masking strategy.***
>
> In Figure 2, We applied various mask ratios to the images and used these two models to reconstruct the masked areas during inference. **When the mask ratio reached 50%, the EDT-S model using MDT's masking strategy struggled to reconstruct the images, while the EDT-S model using EDT's masking strategy was still able to do so. EDT trained by EDT's masking training strategy demonstrates better image reconstruction capabilities.** This visual analysis demonstrates the effectiveness of the EDT's masking training strategy.
>
> ---
>
> ***3.The question for the reported FID on CelebA-HQ.***
>
> Due to the limited time and resources, we train 100k steps for both baselines and our proposed EDT for a fair comparison, using the standard training settings. We agree that the FID is not very promising for the optimal result, but the result still can show better performance than the other baseline methods. We will perform a more steps (400k) training experiment for future revision.
>
> ---
>
> ***4.High resolution result.***
>
> Indeed, the high-resolution result makes the work more convincing. Currently, we are performing the experiment on the Imagenet 512x512, and we will provide the result in the revision.

---

### Author Rebuttal · Authors · 2024-08-07

We thank all the reviewers for their constructive comments and insightful suggestions. We carefully add experiments and figures according to the comments of all the reviewers.

---

We are encouraged that the reviewers pointed out our work
*"is novel"*, *"tackling an important application"* (**R FhNv**);
*"well formulated and clearly explained"*, *"unique contributions"*,  and *"is noteworthy"* (**R QU5R**);
*"reduce the computation of diffusion"* and *"experimental results show promising looks"* (**R 8pvq**);
*"a novel way"*, *"pipeline is robust and lightweight"*, and *"provides specific quantitative results"* (**R MD5e**).
We address the reviews below and will incorporate all changes in the revision.

---

***Summary and Responses to the Reviewers' Most Concerned Issues:***

---

***1. The performance improvements by AMM.***

- We integrated AMM into pre-trained EDT, DiT, and MDT, which are transformer-based models. We conducted experiments on ImageNet and Celeba-HQ **in Rebuttal-Table 1**. **The models with AMM obtain lower FID compared to the models without AMM**, which further demonstrates the effectiveness of AMM in different models and datasets.

|**On ImageNet 256×256 (class-conditional image synthesis)**||||
|---|---|---|---|
|Model|Training images|W/o AMM|W AMM|
|EDT-S|400k * 256|42.60|**34.20**|
|DiT-S|400k * 256|67.16|**63.11**|
|MDTv2-S|400k * 256|39.02|**31.89**|
|EDT-XL|400k * 256|12.80|**7.52**|
|DiT-XL|400k * 256|18.48|**14.73**|
|DiT-XL|7000k * 256|9.62|**3.75**|
|**On CelebA-HQ 256 × 256 (unconditional image synthesis)**||||
|Model|Training images|W/o AMM |W AMM|
|EDT-S|100k * 240|17.01|**16.60**|
|DiT-S|100k * 240|19.12|**18.41**|
|MDTv2-S|100k * 240|17.34|**17.11**|

**Rebuttal-Table 1**: Evaluation of the performance of pre-trained EDT, DiT, and MDT without and with AMM on ImageNet and CelebA-HQ.

---

***2. Experiments on extra dataset CelebA-HQ 256×256.***

- CelebA-HQ 256×256 is used for the unconditional image synthesis task. We conducted a new experiment, training EDT-S, DiT-S, and MDTv2-S on CelebA-HQ 256×256, **as shown in Rebuttal-Table 2 in the global rebuttal. EDT-S achieved the lowest FID, demonstrating that EDT is also effective in unconditional image synthesis tasks.**

|Model|Training images|FID|
|---|---|---|
|EDT-S|100k * 240|**16.60**|
|DiT-S|100k * 240|19.12|
|MDTv2-S|100k * 240|17.34|

**Rebuttal-Table 2**: Evaluation of unconditional image synthesis on CelebA-HQ.

---

***3. The training cost of EDT.***

- For ImageNet 256×256, we estimated the training cost for EDT, MDTv2, and DiT on a 48GB-L40 GPU **in Rebuttal-Table 3 in the global rebuttal**, using a batch size of 256 and FP32 precision. **EDT achieves the best performance with a low training cost.** (GPU days refer to the number of days required for training on a single L40 GPU.)

| Model|Epochs|Training images|GPU days| FID|
|---|---|---|---|---|
|**EDT-S** |80|102M|**2.75**|**38.73**|
|DiT-S|80|102M|2.96|68.40|
|MDTv2-S|80|102M|16.47|39.50|
|**EDT-B**|80|102M|9.19|**19.18**|
|DiT-B|80|102M|**8.62**|43.47|
|MDTv2-B|80|102M|26.17|19.55|
|**EDT-XL**|80|102M|**37.79**|**7.52**|
|DiT-XL|80|102M|39.82|19.47|
|MDTv2-XL|80|102M|72.62|7.70|

**Rebuttal-Table 3**: The training cost and FID of EDT, DiT, and MDTv2 on ImageNet 256×256 with batch size of 256 and FP32 precision.

---

***4. Analysis of masking training mechanism***

- By observing the loss changes **in Rebuttal-Figure 1 in the PDF of the global rebuttal**, we identified a conflict between $L_{masked}$ (loss when the input consists of the remaining tokens after masking) and $L_{full}$ (loss when the input consists of the full token input) in MDTv2.
We separately applied the masking training strategies of MDTv2 and EDT to train diffusion models and extracted $L_{masked}$ and $L_{full}$ values at the 300k~305k training iterations. **As shown in Rebuttal-Figure 1 in the PDF of the global rebuttal**, we visualized the changes of $L_{masked}$ and $L_{full}$. **The left-top of Rebuttal-Figure 1** depicts the loss changes when using MDTv2's masking training strategy. As $L_{full}$ decreases, $L_{masked}$ increases, and vice versa, illustrating the conflict between these two losses. This conflict arises because $L_{masked}$ causes the model to focus on masked token reconstruction while ignoring diffusion training. **As shown in the bottom-left of Rebuttal-Figure 1, both the $L_{full}$ and $L_{masked}$ hardly converged during the 300k to 400k training iterations.**
**The right side of Rebuttal-Figure 1** shows the loss changes when using EDT's masking training strategy. With EDT's strategy, $L_{masked}$ and $L_{full}$ exhibit synchronized changes, and **the loss values continuously decrease during the 300k to 400k training iterations.**

---

***5.  The computational process of the AMM in a block.***

- **In Rebuttal-Figure 2 in the PDF of global rebuttal**, we illustrate the process of attention modulation. Image can be split into N patches and each token is the feature of a patch.  Each token (patch) corresponds to a rectangular area of the image and has a corresponding 2-D coordinate (x, y) in the image. We calculate an Euclidean distance value $d$ for each pair of tokens, resulting in a distance matrix, which is an N×N tensor. Based on the distance matrix, we generate modulation values $m$ via the modulation matrix generation function $F(d)$, which assigns lower modulation values to tokens that are farther apart. These modulation values form an Attention Modulation Matrix (AMM), another N×N tensor. **Importantly, we integrate the AMM into the pre-trained EDT without any additional training.** The attention modulation matrix is calculated when the model is instantiated. During inference, the modulated attention score matrix is obtained by performing a Hadamard product between the attention modulation matrix and the attention score matrix.

---

### Decision · Program_Chairs · 2024-09-25

**Decision:**

Accept (poster)

**Comment:**

Overall, the reviews for this paper are generally positive, with three borderline accepts and one weak accept. The reviewers commend the work for addressing a challenging issue in diffusion, introducing novel and robust techniques, and providing substantial experimental evidence. The rebuttal successfully clarified several points of concern, further convincing the reviewers to some extent. The AC agrees with the reviewers’ assessment and recommends accepting this paper.

However, the reviewers have raised some concerns that should be addressed in the camera-ready version:

1. Provide more qualitative analysis within the main paper rather than in the appendix, including high-resolution comparisons with other methods.
2. Include the experiment of EDT_XL for full convergence and its discussion.
3. Incorporate the additional clarifications and evidence provided in the rebuttal into the final version.

We look forward to seeing this paper presented at NeurIPS.